# hnRNP R promotes O-GlcNAcylation of eIF4G and facilitates axonal protein synthesis

Abdolhossein Zare [1,6], Saeede Salehi [1,6], Jakob Bader [2,6], Cornelius Schneider[3], Utz Fischer [3], Alexander Veh [1], Panagiota Arampatzi [4], Matthias Mann [2,5], Michael Briese [1,7] ✉ & Michael Sendtner [1,7] ✉

Motoneurons critically depend on precise spatial and temporal control of translation for axon growth and the establishment and maintenance of neuromuscular connections. While defects in local translation have been implicated in the pathogenesis of motoneuron disorders, little is known about the mechanisms regulating axonal protein synthesis. Here, we report that motoneurons derived from *Hnrnpr* knockout mice show reduced axon growth accompanied by lowered synthesis of cytoskeletal and synaptic components in axons. Mutant mice display denervated neuromuscular junctions and impaired motor behavior. In axons, hnRNP R is a component of translation initiation complexes and, through interaction with O-linked β-N-acetylglucosamine (O-GlcNAc) transferase (Ogt), modulates O-GlcNAcylation of eIF4G. Restoring axonal O-GlcNAc levels rescued local protein synthesis and axon growth defects of hnRNP R knockout motoneurons. Together, these findings demonstrate a function of hnRNP R in controlling the local production of key factors required for axon growth and formation of neuromuscular innervations.

The development and maintenance of neurons are intricately linked to RNA metabolism. Particularly, mechanisms regulating post-transcriptional RNA processing have been implicated at nearly every step of neuronal differentiation and maturation. Axon growth involves the localization and local translation of mRNAs in axons and growth cones to generate a local supply of cytoskeletal components and other proteins required for axon development[1]. Growth cone turning towards chemo-attractive signals involves local synthesis of β-actin[2]. In adult mouse brains, ribosomes are present in presynaptic compartments, contributing to the diversity of the local proteome[3–5]. Furthermore, there is evidence that ribosomes are locally remodeled in axons to fine-tune local protein production[6,7].

Axonal mRNA localization and local translation are tightly controlled by RNA-binding proteins (RBPs), which are not only abundant in the nucleus of neurons where they control splicing and polyadenylation, but are often also transported into axons as part of messenger ribonucleoprotein particles (mRNPs)[8]. In line with this notion, hundreds of RNAs have been detected in axons of motoneurons and other neuronal subtypes[9,10]. While the functions of RBPs in axonal RNA transport have been studied in detail, the mechanisms through which they regulate local mRNA translation are not yet well understood. Local protein synthesis is particularly important for motoneurons as they send out long and highly branched axons to connect to their target muscles. Accordingly, defective local translation is an early event in the pathogenesis underlying motoneuron disorders such as amyotrophic lateral sclerosis (ALS), contributing to synapse loss and motoneuron degeneration[11].

[1]Institute of Clinical Neurobiology, University Hospital Wuerzburg, Wuerzburg, Germany. [2]Department of Proteomics and Signal Transduction, Max Planck Institute of Biochemistry, Martinsried, Germany. [3]Department of Biochemistry, Theodor Boveri Institute, University of Wuerzburg, Wuerzburg, Germany. [4]Core Unit Systems Medicine, University of Wuerzburg, Wuerzburg, Germany. [5]NNF Center for Protein Research, Faculty of Health Sciences, University of Copenhagen, Copenhagen, Denmark. [6]These authors contributed equally: Abdolhossein Zare, Saeede Salehi, Jakob Bader. [7]These authors jointly supervised this work: Michael Briese, Michael Sendtner. ✉e-mail: Briese_M@ukw.de; Sendtner_M@ukw.de

The RBP heterogeneous nuclear ribonucleoprotein R (hnRNP R) is abundant in the nervous system and highly expressed during the period of neuronal development[12]. In motoneurons cultured from embryonic mouse spinal cords, hnRNP R is present at high levels in the nucleus but also detectable in the cytosol including axons and growth cones[12]. In agreement, hnRNP R has also been observed in presynaptic regions of neuromuscular junctions (NMJs) in vivo[13]. Depletion of hnRNP R in cultured motoneurons and in developing zebrafish via knockdown resulted in reduced axon growth while survival of neurons was unaffected[14]. hnRNP R was found to associate with the 3'UTR of the mRNA encoding β-actin and depletion of hnRNP R affected the localization of β-actin mRNA into axons[15]. Additional RNA interactors of hnRNP R in motoneurons were subsequently identified by individual-nucleotide resolution crosslinking and immunoprecipitation (iCLIP) and the axonal transport of many of them was perturbed following hnRNP R loss[16]. Among these was the non-coding RNA 7SK, an abundant nuclear RNA regulating transcription, which, in motoneurons, is also present in axons. Thus, hnRNP R shapes the composition of the axonal transcriptome during axon growth.

While functions of hnRNP R in axon growth and axonal RNA transport have been elucidated in cultured motoneurons, its role for motoneuron development in vivo is not understood yet. Here, we investigated the consequences of hnRNP R loss on motoneuron development using an *Hnrnpr* knockout mouse. These *Hnrnpr* knockout mice showed reduced motor functions accompanied by impaired motoneuron innervation of neuromuscular junctions. In agreement with the hnRNP R knockdown phenotype, motoneurons cultured from *Hnrnpr* knockout mice had reduced axon growth while their survival was unaffected. Surprisingly, we observed decreased protein synthesis selectively in axons of hnRNP R-deficient motoneurons, accompanied by reduced axonal localization of the translational machinery. hnRNP R itself interacted with ribosomes and translation factors and was required for translation initiation in axons.

Proteomic analysis of axons of hnRNP R knockout motoneurons revealed a number of cytoskeletal proteins, synaptic proteins, ribosomal proteins and translation factors whose axonal levels were particularly dependent on hnRNP R. We identified O-linked β-N-acetylglucosamine (O-GlcNAc) transferase (Ogt) as a major hnRNP R interactor in axons and showed that hnRNP R regulates O-GlcNAcylation of the essential translation initiation factor eIF4G. Together, our results indicate that hnRNP R positively regulates translation in axons of motoneurons, thereby facilitating axon growth and neuromuscular connectivity.

## Results

### Generation and characterization of *Hnrnpr* knockout mice

Previous studies have shown that hnRNP R modulates axon growth in motoneurons[14,16,17]. To explore the functional importance of hnRNP R in motoneuron development in vivo, we sought to generate an *Hnrnpr* knockout mouse model. The *Hnrnpr* pre-mRNA undergoes alternative splicing of exon 2, giving rise to two hnRNP R protein isoforms that differ at the N-terminus due to AUG start codons present in exons 2 and 4[18]. Therefore, to generate full *Hnrnpr* knockout mice, we used CRISPR/Cas9-mediated deletion of a genomic region encompassing *Hnrnpr* exons 1 to 5 (Supplementary Fig. 1a). PCR genotyping was performed using three primers (Supplementary Fig. 1a, b). The three genotypes (+/+, +/−, −/−) occurred according to the expected Mendelian ratio and survival of *Hnrnpr*$^{-/-}$ mice up to 18 months was unaffected (Fig. 1a). In *Hnrnpr*$^{-/-}$ mice, *Hnrnpr* transcript levels were undetectable by qPCR (Supplementary Fig. 1c). Immunoblot analysis using an antibody targeting the hnRNP R C-terminus confirmed loss of both protein isoforms in *Hnrnpr*$^{-/-}$ mice (Supplementary Fig. 1d, e). Depletion of hnRNP R protein in *Hnrnpr*$^{-/-}$ mice was also confirmed using antibodies targeting the N-terminal of hnRNP R (Supplementary Fig. 1f, g). Next, we cultured primary embryonic motoneurons from *Hnrnpr*$^{-/-}$ and +/+ mice for 6 days in vitro (DIV) (Fig. 1b). *Hnrnpr*$^{-/-}$

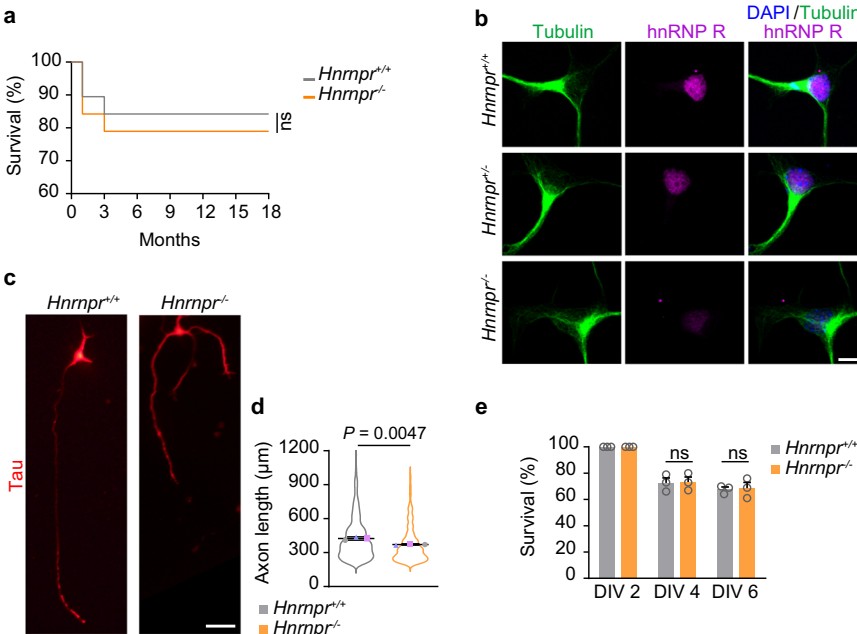

**Fig. 1 | Generation and validation of *Hnrnpr* knockout mice. a** Survival analysis of *Hnrnpr* mutant mice. (Mean survival after 18 months: *Hnrnpr*$^{+/+}$: 83.3% $n$ = 22 mice; *Hnrnpr*$^{-/-}$: 81.8%, $n$ = 18 mice). Statistical analysis was performed using a Log-rank (Mantel-Cox) test. **b** Representative images of hnRNP R immunofluorescence in cultured motoneurons at DIV 6. Scale bar, 10 μm. The images are representative of three biological replicates. **c** Morphology of cultured DIV 6 motoneurons from *Hnrnpr*$^{+/+}$ and *Hnrnpr*$^{-/-}$ mice immunostained with an antibody against tau. Scale

bar, 50 μm. **d** SuperPlots of axon lengths of motoneurons. Statistical analysis was performed using an unpaired two-tailed Student's $t$ test. Data are mean ± s.d. of three biological replicates. *Hnrnpr*$^{+/+}$, $n$ = 629; *Hnrnpr*$^{-/-}$, $n$ = 477 motoneurons.
**e** Quantification of motoneuron survival. Statistical analysis was performed using a two-way ANOVA with Tukey's multiple comparisons test. ns, not significant. Data are mean ± s.d. of three biological replicates. Source data are provided as a Source Data file.

motoneurons displayed reduced axon lengths (Fig. 1c, d and Supplementary Fig. 1h), while survival was unaffected (Fig. 1e). These findings suggest that hnRNP R exerts functions during axon growth in developing motoneurons.

## hnRNP R deficiency disturbs the axonal proteome

Guided by our previous proteomics data showing the association of hnRNP R with translation factors and ribosomal proteins[18], we investigated the possibility that hnRNP R regulates axonal protein synthesis. For this purpose, we pulsed primary motoneurons cultured from *Hnrnpr*$^{-/-}$ and +/+ mice with puromycin, which gets incorporated into newly synthesized proteins. We observed a significant reduction in the axonal puromycin immunosignal, both in proximal and distal regions, in *Hnrnpr*$^{-/-}$ motoneurons, whereas puromycin incorporation in somata was unchanged (Fig. 2a, b). Puromycin immunolabelling was strongly reduced when motoneurons were pre-treated with the protein synthesis inhibitor cycloheximide or when puromycin was omitted indicating the specificity of the labeling procedure (Supplementary Fig. 2a). Thus, hnRNP R deficiency reduces axonal protein synthesis.

Next, we sought to identify axonal proteins altered in *Hnrnpr*$^{-/-}$ motoneurons. To do so, we cultured *Hnrnpr*$^{-/-}$ and +/+ motoneurons for 7 DIV in compartmentalized microfluidic chambers allowing the separation of axons from the somatodendritic compartment (Fig. 2c, d). Total protein was extracted from both compartments (Supplementary Fig. 2b, c) and subjected to mass spectrometry-based analysis (Supplementary datasets 1–3). Comparison of axonal and somatodendritic protein levels in wildtype motoneurons revealed an enrichment of the axonal marker proteins tau (Mapt), β-actin (Actb), and Tubb3 in the axonal proteome (Fig. 2e; Supplementary dataset 1). In contrast, nuclear markers such as RNA polymerase subunits and dendritic proteins such as Map2 were enriched in the somatodendritic proteome (Fig. 2e; Supplementary dataset 1). Comparison of the axonal proteomes of *Hnrnpr*$^{-/-}$ and +/+ motoneurons revealed that the levels of 344 and 125 proteins were downregulated and upregulated, respectively, by hnRNP R deficiency (Fig. 2f; Supplementary dataset 3). Among the downregulated axonal proteins, 53 were also downregulated in the somatodendritic compartment and, among the upregulated axonal proteins, 7 were also upregulated in the somatodendritic compartment of *Hnrnpr*$^{-/-}$ motoneurons (Fig. 2g; Supplementary datasets 2, 3). Thus, only a subset of protein alterations occurring in axons of hnRNP R-deficient motoneurons is a consequence of changes in their total levels.

Proteins dysregulated in the axonal compartment of *Hnrnpr*$^{-/-}$ motoneurons were associated with synaptic functions (Fig. 2h, i and Supplementary Fig. 2d). Additionally, while we did not observe a general downregulation of proteins with cytoskeletal functions, we identified several proteins with known cytoskeletal roles in axon development and maintenance such as Stmn2, Stmn4, Tuba1a, Tubb2a, and Tubb4a that were reduced in axons but not somatodendritic regions of *Hnrnpr*$^{-/-}$ motoneurons (Fig. 2h, i 'custom cytoskeleton'; Supplementary dataset 4). In the somatodendritic compartment, we detected 95 ribosomal proteins, of which 16 and 11 ribosomal proteins were upregulated and downregulated, respectively, in hnRNP R-depleted motoneurons (Supplementary Fig. 2e; Supplementary dataset 2). In the axonal compartment, 61 ribosomal proteins were detected, of which two ribosomal proteins (eS2 and eL13) were upregulated, and 5 ribosomal proteins (eS18, eS3, eL14, eL15, and eS15a) were downregulated upon loss of hnRNP R (Supplementary Fig. 2e; Supplementary datasets 2, 3). We found that, among deregulated ribosomal proteins, only eS2 was elevated in both compartments. Therefore, based on these data, we did not detect a general depletion of ribosomes in axons of *Hnrnpr*$^{-/-}$ motoneurons.

The most strongly downregulated axonal protein was microtubule actin crosslinking factor 1 (Macf1) protein that plays a role in

cytoskeletal dynamics[19–21] (Supplementary dataset 3). We investigated its local synthesis in axons of *Hnrnpr*$^{-/-}$ motoneurons using the puromycylation followed by proximity ligation amplification assay (Puro-PLA)[22] (Fig. 3a). We observed that the Puro-PLA signal for Macf1 was significantly reduced in axons but not somata of hnRNP R-deficient motoneurons (Fig. 3b). No Puro-PLA signal was detected when puromycin was omitted indicating the specificity of the labeling procedure (Fig. 3a). To further evaluate the effects of hnRNP R depletion on axonal Macf1 translation, we immunoprecipitated ribosomes with the Y10b antibody from somatodendritic and axonal lysates of compartmentalized *Hnrnpr*$^{-/-}$ and +/+ motoneurons and assessed *Macf1* mRNA co-purification by qPCR. In axons of hnRNP R-depleted motoneurons, the association of *Macf1* mRNA with ribosomes was strongly reduced while it remained unchanged in the somatodendritic compartment (Fig. 3c). As control, the total levels of *Macf1* mRNA were unchanged in the axonal compartment of *Hnrnpr*$^{-/-}$ motoneurons indicating that the reduced association of *Macf1* with ribosomes in axons is not due to lowered mRNA levels (Fig. 3d). Together, these data indicate that hnRNP R depletion dysregulates proteins of several pathways involved in axon growth including cytoskeleton assembly and protein synthesis.

## Axonal proteome alterations of hnRNP R knockout motoneurons are not due to altered levels of axonal mRNA

In order to assess whether dysregulation of axonal proteins in *Hnrnpr* knockout motoneurons is due to the altered axonal level of encoding transcripts, we investigated changes in the somatodendritic and axonal mRNA levels in compartmentalized *Hnrnpr*$^{-/-}$ and +/+ motoneurons by RNA sequencing (RNA-seq) (Supplementary datasets 5, 6). To assess data quality, we first performed differential expression analysis comparing somatodendritic with axonal mRNA levels of *Hnrnpr*$^{+/+}$ motoneurons. We compared the log$_2$ fold changes with those obtained by our previous RNA-seq analysis of compartmentalized wildtype motoneurons[9]. Both datasets were strongly correlated (Pearson's $r = 0.9$) indicating reproducibility (Fig. 4a). Next, we investigated axonal transcript changes in *Hnrnpr*$^{-/-}$ compared to +/+ motoneurons by differential expression analysis. Using an adjusted $p$-value (Padj.) < 0.05 as cutoff revealed two transcripts (*Gm15477* and *Stra6*) that were upregulated and three transcripts (*Hnrnpr*, *Gm15477*, and *LOC118568625*) that were downregulated in axons of *Hnrnpr*$^{-/-}$ motoneurons (Fig. 4b). In the somatodendritic compartment of *Hnrnpr*$^{-/-}$ motoneurons, no transcripts were upregulated and downregulated (Fig. 4b). Using a less stringent cutoff of $P < 0.05$, we identified 1194 dysregulated axonal transcripts, of which 412 transcripts were upregulated and 782 transcripts were downregulated, and 198 dysregulated somatodendritic transcripts, of which 112 were upregulated and 86 were downregulated (Fig. 4c). Among these, 5 and 3 transcripts were up- or downregulated, respectively, in both somatodendritic and axonal compartments of *Hnrnpr*$^{-/-}$ motoneurons (Fig. 4d). To assess hnRNP R binding to dysregulated transcripts, we overlaid our RNA-seq data with a previously published hnRNP R iCLIP datasets from embryonic mouse motoneurons[16]. We observed that transcripts downregulated in axonal compartments of *Hnrnpr*$^{-/-}$ motoneurons contain more hnRNP R iCLIP hits whereas upregulated transcripts in both somatodendritic and axonal compartments contain less hnRNP R iCLIP hits compared to unchanged transcripts (Fig. 4e).

To investigate whether axonal proteins dysregulated in axons of *Hnrnpr*$^{-/-}$ motoneurons are changed at the transcript level, we cross-compared our axonal RNA-seq and proteomics datasets. Our analysis demonstrated that only seven dysregulated axonal transcripts were also changed at the protein level ($P < 0.05$ in both) (Fig. 4f, g). This suggests that the protein changes detected in axons of hnRNP R-deficient motoneurons are mostly not a consequence of transcript alterations but rather might be due to impaired local protein synthesis.

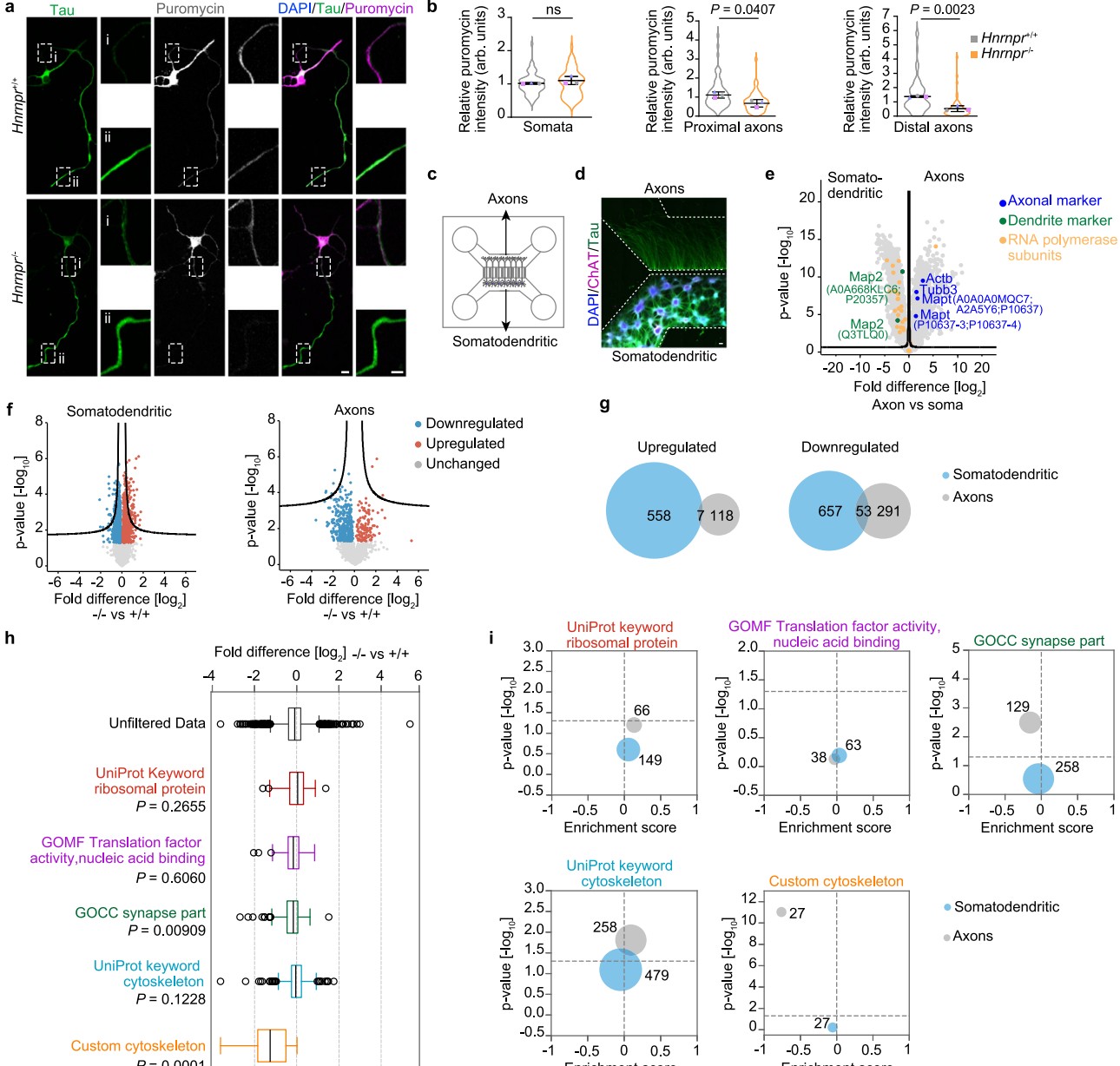

**Fig. 2 | Axonal protein synthesis is impaired in *Hnrnpr*⁻/⁻ motoneurons.**
**a** Immunostaining of DIV 6 puromycin-pulsed motoneurons cultured from *Hnrnpr*⁻/⁻ and +/+ mice, with proximal and distal regions of the axon marked. Scale bars, 10 μm and 5 μm (magnified areas). **b** SuperPlots of puromycin immunosignals in the somata and proximal and distal axonal regions of *Hnrnpr*⁻/⁻ and +/+ motoneurons. Statistical analysis was performed using an unpaired two-tailed Student's *t* test. Data are mean ± s.d. of three biological replicates. *Hnrnpr*⁺/⁺, *n* = 69; *Hnrnpr*⁻/⁻, *n* = 60 motoneurons. **c** Schematic of a microfluidic chamber for compartmentalized motoneuron cultures allowing separation of the somatodendritic from axonal compartments. **d** Immunostaining of DIV 7 motoneurons cultured in a microfluidic chamber. Scale bar, 100 μm. The images are representative of at least three biological replicates. **e** Volcano plot of differential protein abundance in the axonal compared to the somatodendritic compartment of wildtype motoneurons cultured

in microfluidic chambers from four biological replicates. Statistical analysis was performed using an unpaired two-sided Student's *t* test. **f** Volcano plots of proteins differentially enriched in axonal and somatodendritic compartments of *Hnrnpr*⁻/⁻ compared to +/+ motoneurons cultured in microfluidic chambers from four biological replicates. Statistical analysis was performed using an unpaired two-sided Student's *t* test. **g** Overlap of proteins in (**f**) significantly (*P* < 0.05) deregulated in the somatodendritic and axonal compartments of hnRNP R knockout motoneurons. **h** Box plots highlighting differential enrichment of ribosomal proteins, translation factors, synaptic proteins and cytoskeletal components in axons of *Hnrnpr*⁻/⁻ relative to +/+ motoneurons cultured in microfluidic chambers from four biological replicates. Statistical analysis was performed using Welch's *t* test. **i**, Enrichment scores of protein subgroups in axons of *Hnrnpr*⁻/⁻ relative to +/+ motoneurons. The size of the circles represents the number of proteins.

## hnRNP R associates with the translation initiation machinery

Having shown that axonal translation is impaired in hnRNP R-deficient motoneurons, we asked whether hnRNP R might have an as yet unknown function in regulating axonal translation by acting as a ribosome-associated protein. To test this possibility, we used the Y10b antibody to immunoprecipitate ribosomes from motoneuron lysate, or from lysate of motoneuron-like NSC-34 cells[23]. We observed co-

purification of hnRNP R and eS5 (as positive control), but not Gapdh (as negative control), with ribosomes immunoprecipitated with Y10b (Fig. 5a). Following RNase I treatment of lysates to digest ribosome-associated mRNAs but not ribosomal RNA (rRNA), the association of hnRNP R with ribosomes was abolished (Fig. 5a). To investigate the association of hnRNP R with ribosomes specifically in axons, we cultured motoneurons in microfluidic chambers (Fig. 5b, c) and

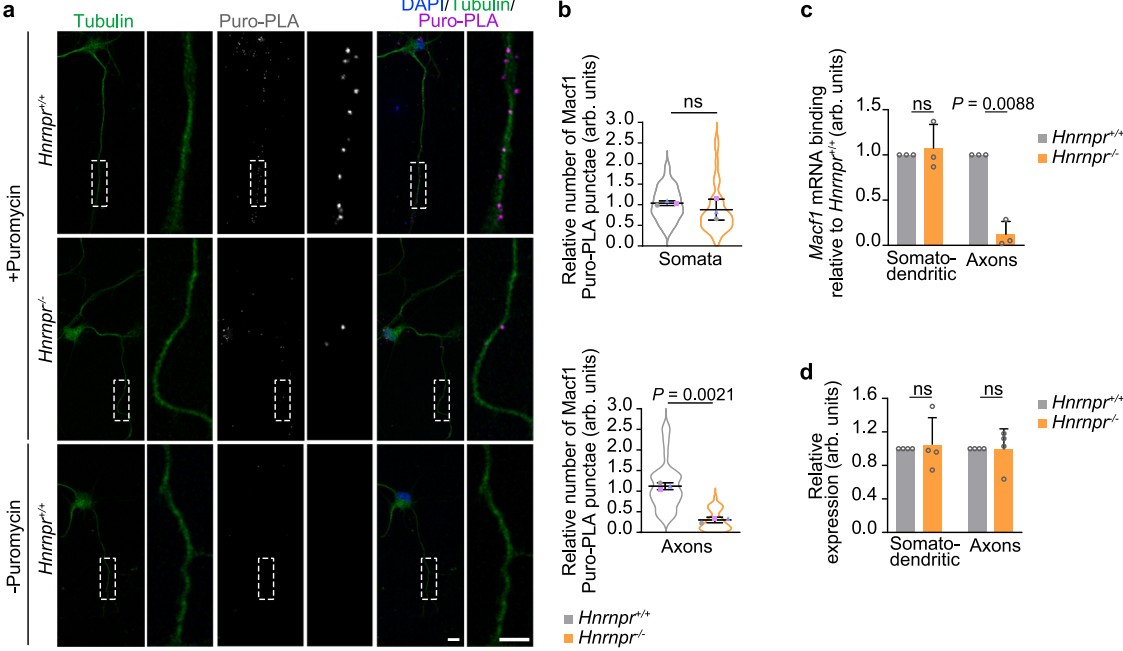

**Fig. 3 | Axonal Macf1 protein synthesis is regulated by hnRNP R.**
**a** Representative images of Macf1 Puro-PLA signal in cultured DIV 6 *Hnrnpr*⁻/⁻ and +/+ motoneurons using anti-puromycin and anti-Macf1 antibodies. Labeling without puromycin was used as a control. Scale bars, 10 μm, and 5 μm (magnified areas). **b** SuperPlots of the relative number of Macf1 Puro-PLA punctae in somata and 50 μm of axons of cultured *Hnrnpr*⁻/⁻ and +/+ motoneurons. Unpaired two-tailed Student's *t* test. Data are mean ± s.d. of three biological replicates. *Hnrnpr*⁺/⁺, *n* = 37; *Hnrnpr*⁻/⁻, *n* = 38 motoneurons. **c** qPCR analysis of *Macf1* mRNA co-precipitated by

Y10b from *Hnrnpr*⁻/⁻ and +/+ motoneurons cultured in a compartmentalized microfluidic chamber. Statistical analysis was performed using a two-tailed one-sample *t* test. Data are mean ± s.d. of three biological replicates. **d** Quantification of the total levels of *Macf1* mRNA in the somatodendritic and axonal compartments of *Hnrnpr*⁻/⁻ and +/+ motoneurons cultured in compartmentalized microfluidic chambers. Gapdh was used for normalization. Statistical analysis was performed using a two-tailed one-sample t-test. Data are mean ± s.d. of four biological replicates. Source data are provided as a Source Data file.

performed sucrose density gradient ultracentrifugation of somato-dendritic and axonal lysate to separate polysomes from 80S mono-somes and 40S and 60S ribosomal subunits (Fig. 5d). We observed that hnRNP R was detectable at higher level in the 40S, 60S, and monosome fractions compared to the polysome fractions in both somatodendritic and axonal compartments, using eIF2α as a control for the distribution of ribosomal subunits and monosomes across the fractions (Fig. 5e). Additionally, hnRNP R was present in the light fractions containing mRNPs in lysates of both compartments in agreement with its role as RNA binding protein. To a lesser extent, hnRNP R was also detectable in polysome fractions of the somatodendritic lysate (Fig. 5e). Distribution of 18S rRNA was detected across the gradient in both somatodendritic and axonal lysate by qPCR, with a peak in fractions containing monosomes. For axonal lysate, an additional peak in 18S rRNA levels was detectable in fractions 10 and 11, most likely representing translating polysomes (Fig. 5f). Together, our data indicate that hnRNP R associates with ribosomes at the translation initiation stage in axons of motoneurons.

To substantiate our finding that hnRNP R interacts with initiating ribosomes, we immunoprecipitated hnRNP R from motoneurons lysate and observed that eIF2α, but not the elongation factor eEF2, co-purified with hnRNP R (Fig. 5g). Following treatment of lysate with Benzonase to digest mRNA as well as rRNA, co-purification of eIF2α with hnRNP R was unperturbed showing that their association is RNA-independent (Fig. 5g). To confirm the association between hnRNP R and eIF2α in situ, we performed PLA using antibodies against hnRNP R and eIF2α. We observed that the hnRNP R-eIF2α PLA signal was detectable not only in the cytosol of the somata but also in axons and growth cones of motoneurons (Fig. 5h). No PLA signals were detectable when either hnRNP R or eIF2α antibody was omitted indicating specificity of the procedure (Supplementary Fig. 3a). In addition to eIF2α, the initiation factor eIF4G, a component of the eIF4F cap-

interacting complex, was co-precipitated by anti-hnRNP R in an RNA-independent manner (Fig. 5g). Together, these results point towards the possibility that hnRNP R regulates translation at the initiation stage.

## Axonal translation initiation in motoneurons is dependent on hnRNP R

To investigate whether the loss of hnRNP R leads to impaired axonal translation initiation, we monitored the association between cyto-plasmic polyadenylate-binding protein 1 (Pabpc1; hereafter referred to as Pabp) and eIF4G. According to the closed-loop model, close proximity between eIF4G and Pabp is an essential step in eukaryotic translation[24]. We assessed the interaction between Pabp and eIF4G in situ by applying PLA to cultured *Hnrnpr*⁻/⁻ and +/+ motoneurons using antibodies against Pabp and eIF4G (Fig. 6a and Supplementary Fig. 3b). We observed a significantly reduced Pabp-eIF4G PLA signal in axons but not somata of hnRNP R-deficient motoneurons (Fig. 6b) indicative of reduced translation initiation. To validate our finding, we performed PLA in cultured *Hnrnpr*⁻/⁻ and +/+ motoneurons using antibodies against the ribosomal proteins eS6 and eL24 to evaluate ribosome assembly[25] (Fig. 6c and Supplementary Fig. 3c). eS6 and eL24 are components of the 40S and 60S ribosomal subunits, respectively, and form an intersubunit bridge that is critical for 80S ribosome assembly[26]. In agreement with our Pabp-eIF4G PLA analysis, we detected a significantly reduced eL24-eS6 PLA signal in axons but not somata of *Hnrnpr*⁻/⁻ motoneurons (Fig. 6d), further supporting the notion of reduced axonal ribosome assembly upon hnRNP R deficiency. Failure to assemble might lead to the degradation of ribo-somes. To investigate ribosome levels, we immunostained *Hnrnpr*⁻/⁻ and +/+ motoneurons with Y10b (Fig. 6e). Y10b immunosignal intensity was significantly reduced in axons of hnRNP R-deficient motoneurons while it remained unchanged in somata (Fig. 6f). Together, our findings

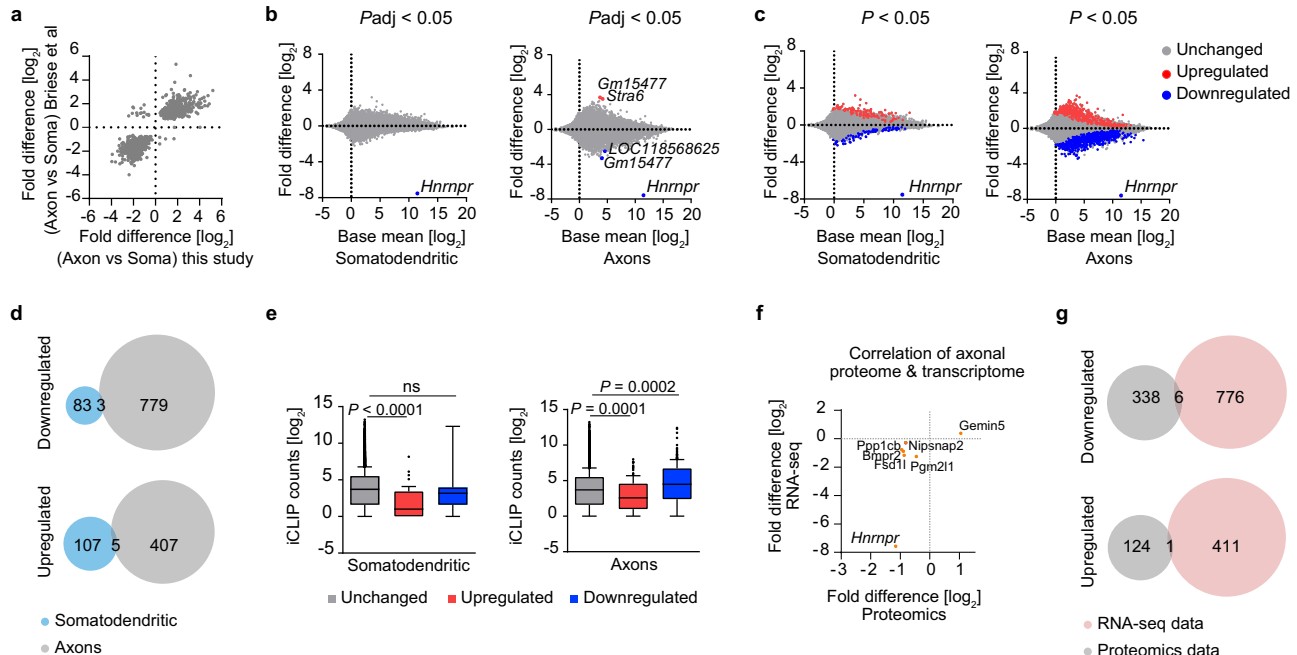

**Fig. 4 | Axonal transcriptome alterations of hnRNP R knockout motoneurons are distinct from proteome changes. a** Scatter plot correlating logarithmized fold changes of transcripts differentially expressed (*P*adj < 0.05) in the axonal relative to the somatodendritic compartment of wildtype motoneurons from Briese et al.[9] with this study. **b**, **c** Mean average (MA) plots showing somatodendritic and axonal transcriptome changes in *Hnrnpr*⁻/⁻ relative to +/+ motoneurons cultured in microfluidic chambers from four biological replicates. Transcripts altered with *P*adj < 0.05 (**b**) or *P* < 0.05 (**c**) are highlighted. **d** Venn diagrams showing the overlap of deregulated (*P* < 0.05) transcripts in the somatodendritic and axonal compartments of *Hnrnpr*⁻/⁻ motoneurons. **e** Box plots showing the number of iCLIP hits in transcripts differentially expressed in axons of *Hnrnpr*⁻/⁻ relative to +/+ motoneurons cultured in microfluidic chambers from four biological replicates. Statistical analysis was performed using a Kruskal-Wallis test with Dunn's multiple comparisons test. **f** Scatter plot correlating logarithmized fold changes of proteins and transcripts differentially abundant (*P* < 0.05) in axons of *Hnrnpr*⁻/⁻ relative to +/+ motoneurons. **g** Venn diagrams showing the overlap of deregulated (*P* < 0.05) proteins with transcripts deregulated (*P* < 0.05) in the axonal compartment of *Hnrnpr*⁻/⁻ relative to +/+ motoneurons.

indicate that hnRNP R promote axonal translation in motoneurons through facilitating the assembly of translation initiation complexes.

## Ogt is a major interactor of hnRNP R in axons

To gain further insights into the axonal role of hnRNP R in translation initiation, we investigated the axonal protein interactome of hnRNP R in motoneurons. For this purpose, we immunopurified hnRNP R complexes from somatodendritic and axonal compartments of motoneurons cultured in microfluidic chambers using an antibody against the C-terminus of hnRNP R and identified proteins by mass spectrometry. Compared to no antibody control purifications, 275 and 101 proteins were significantly enriched in the hnRNP R immunoprecipitate from the somatodendritic and axonal compartment, respectively (Supplementary datasets 7, 8). Among these, hnRNP R was the top hit indicating the specificity of the procedure (Fig. 7a). We identified O-linked β-N-acetylglucosamine (O-GlcNAc) transferase (Ogt) as the most enriched protein interactor of hnRNP R in axons of motoneurons (Fig. 7a). However, Ogt was not enriched in the hnRNP R immunoprecipitate from the somatodendritic compartment (*P* = 0.153) (Supplementary dataset 7). Among hnRNP R interactors, only 15 proteins associated with hnRNP R in the somatodendritic compartment also interacted with hnRNP R in the axonal compartment further indicating that axonal hnRNP R complexes have a different composition compared to somatodendritic hnRNP R complexes (Fig. 7b). To confirm the specificity of the association between hnRNP R and Ogt in axons, we performed mass spectrometry analysis on Ogt immunoprecipitates from axonal and somatodendritic compartments (Fig. 7c; Supplementary datasets 9, 10). We found that hnRNP R was enriched among Ogt interactors from the axonal but not from the somatodendritic compartment (Fig. 7c), and that axonal Ogt complexes are largely distinct from somatodendritic Ogt complexes

(Fig. 7d), in agreement with the hnRNP R interactome data. When we compared the hnRNP R and Ogt interactome datasets, we observed that the number of proteins common to hnRNP R and Ogt complexes was higher in axons than in somatodendritic regions (Fig. 7e). To validate the association between hnRNP R and Ogt in axons, we immunopurified hnRNP R from somatodendritic and axonal lysate followed by immunoblot analysis. We observed a strong interaction of Ogt with hnRNP R in axons but not in the somatodendritic compartment (Fig. 7f). Additionally, we performed PLA in cultured *Hnrnpr*⁻/⁻ and +/+ motoneurons using antibodies against Ogt and hnRNP R. PLA signal was observed in axons and growth cones of *Hnrnpr*⁺/⁺ motoneurons while it was absent in −/− motoneurons as control (Fig. 7g). Together, these data show that hnRNP R selectively associates with Ogt in axons of motoneurons.

O-GlcNAcylation is a predominant posttranslational modification of cytosolic proteins[27]. To investigate whether hnRNP R regulates Ogt activity in motoneurons, we immunostained *Hnrnpr*⁻/⁻ and +/+ motoneurons with the RL2 antibody against O-GlcNAc (Fig. 7h). Compared to *Hnrnpr*⁺/⁺ motoneurons, O-GlcNAc levels were reduced approximately by half in axons of *Hnrnpr*⁻/⁻ motoneurons whereas somatic O-GlcNAc levels were unchanged (Fig. 7i). In contrast, axonal Ogt levels were not significantly altered in *Hnrnpr*⁻/⁻ relative to +/+ motoneurons (Supplementary Fig. 4a, b). To test whether the reduced axonal O-GlcNAc levels observed in hnRNP R-deficient motoneurons can be restored, we treated *Hnrnpr*⁻/⁻ and +/+ motoneurons with thiamet-G (TMG), an inhibitor of O-GlcNAcase (Oga) that catalyzes the removal O-GlcNAc from modified proteins[28]. We observed that suppression of Oga with TMG restored axonal O-GlcNAc levels in *Hnrnpr*⁻/⁻ motoneurons to those observed in *Hnrnpr*⁺/⁺ motoneurons while Ogt levels were unchanged (Supplementary Fig. 4c–e). Thus, hnRNP R promotes Ogt-mediated O-GlcNAcylation of axonal proteins.

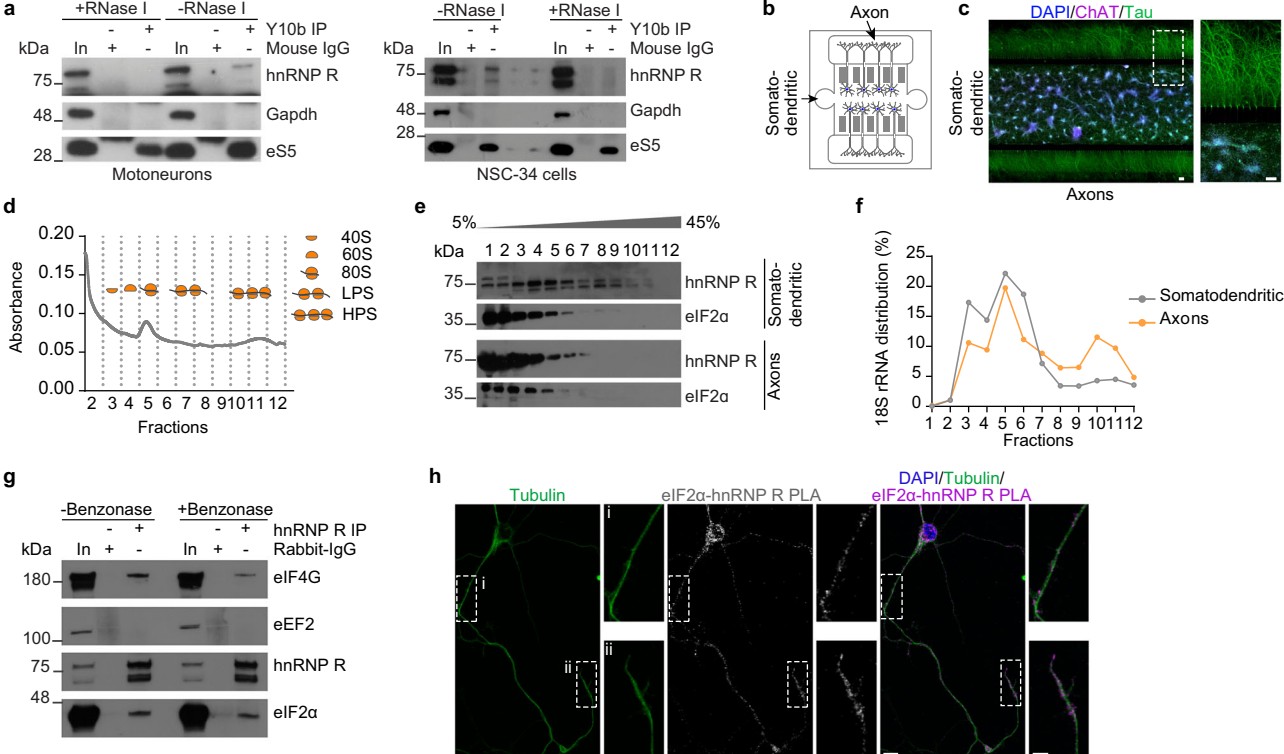

**Fig. 5 | hnRNP R is associated with translation initiation complexes in motoneurons. a** Immunoblot analysis of hnRNP R co-immunoprecipitation by Y10b from motoneuron and NSC-34 cell lysate with or without RNase I treatment. Gapdh and eS5 were used as a negative and positive control, respectively. The immunoblot is representative of two biological replicates. **b** Schematic of a microfluidic chamber for compartmentalized motoneuron cultures allowing separation of the somatodendritic from axonal compartments. **c** Immunostaining of motoneurons cultured in a compartmentalized microfluidic chamber. Scale bars, 100 μm, and 100 μm (magnified areas). The images are representative of at least three biological replicates. **d** Absorbance profile of somatodendritic lysate subjected to sucrose density gradient ultracentrifugation and fractionation. Data are representative of two biological replicates. **e** Immunoblot analysis of hnRNP R and eIF2α in sucrose gradient fractions from somatodendritic and axonal lysates. For fractions 1 and 2 of the somatodendritic compartment, 10% were loaded for analysis. Data are representative of two biological replicates. **f** Quantification of 18S rRNA by qPCR analysis in sucrose gradient fractions from somatodendritic and axonal lysates. Data are representative of two biological replicates. **g** Immunoblot analysis of proteins co-immunoprecipitated by an anti-hnRNP R antibody from motoneuron lysate without or with Benzonase treatment. The immunoblot is representative of two biological replicates. **h** Representative images of hnRNP R-eIF2α PLA signal in cultured motoneurons at DIV 6 using anti-hnRNP R and anti-eIF2α antibodies. Scale bars, 10 μm, and 5 μm (magnified areas). The images are representative of two biological replicates. Source data are provided as a Source Data file.

## hnRNP R is required for Ogt-mediated O-GlcNAcylation of eIF4G

To investigate hnRNP R-dependent O-GlcNAcylation of axonal proteins, we performed immunoprecipitation with the RL2 antibody from axonal lysate of *Hnrnpr*[−/−] motoneurons followed by mass spectrometry analysis of immunoprecipitates. We observed reduced O-GlcNAcylation of eIF4G in axons of *Hnrnpr*[−/−] motoneurons, which could be rescued by TMG treatment (Fig. 8a-c). In contrast, O-GlcNAcylation of axonal eIF4A, another component of the eIF4F complex, was not dependent on hnRNP R. Using immunoblot analysis, we confirmed that eIF4G but not eIF4A is O-GlcNAcylated in axons of motoneurons (Fig. 8d). We then assessed O-GlcNAcylation of eIF4G in cultured *Hnrnpr*[−/−] and +/+ motoneurons by PLA using the RL2 antibody and an antibody against eIF4G (Fig. 8e and Supplementary Fig. 5a). We observed reduced eIF4G O-GlcNAcylation in axons but not somata of *Hnrnpr*[−/−] motoneurons (Fig. 8f). To further validate that the reduction of axonal O-GlcNAcylation of eIF4G in *Hnrnpr*[−/−] motoneurons is the consequence of hnRNP R loss, we transduced motoneurons cultured from *Hnrnpr*[−/−] and +/+ mice with a lentiviral construct expressing an hnRNP R-EGFP fusion protein. Expression of hnRNP R-EGFP could restore the level of O-GlcNAcylation of axonal eIF4G in hnRNP R-depleted motoneurons (Supplementary Fig. 5b, c). Importantly, treatment with TMG could restore O-GlcNAcylation of axonal eIF4G in hnRNP R-deficient motoneurons (Fig. 8e, f).

It has been shown that O-GlcNAcylation of eIF4G on Ser61 by Ogt promotes its interaction with Pabp, thereby stimulating translation initiation[27]. In agreement with this notion, we observed reduced eIF4G association with Pabp by PLA in axons of motoneurons following treatment with the Ogt inhibitor OSMI-1 (Supplementary Fig. 5d, e). In axons of *Hnrnpr*[−/−] motoneurons, we detected a reduced interaction between eIF4G and Pabp, which could be restored by TMG treatment (Fig. 8g, h). Furthermore, TMG treatment could rescue the impaired axonal protein synthesis (Fig. 8i, j) as well as the axon growth defect in hnRNP R-deficient motoneurons (Fig. 8k, l). Similar to TMG treatment, reintroducing hnRNP R could rescue the axon elongation defect observed in *Hnrnpr*[−/−] motoneurons (Supplementary Fig. 5f, g). Thus, while we cannot exclude the possibility that O-GlcNAcylation of additional proteins regulating translation is dependent on hnRNP R, our results indicate that hnRNP R positively regulates eIF4G activity and, thus, the formation of initiation complexes by stimulating Ogt.

## hnRNP R deficiency causes impaired motor functions

Having shown that hnRNP R deficiency affects axon growth and local translation in cultured motoneurons, we investigated whether loss of hnRNP R leads to motor abnormalities in vivo. While survival of *Hnrnpr*[−/−] mice was normal (Fig. 1h) they displayed a significant reduction in body weight compared to +/+ mice (Fig. 9a). *Hnrnpr*[−/−] mice showed reduced motor performance as assessed by grip

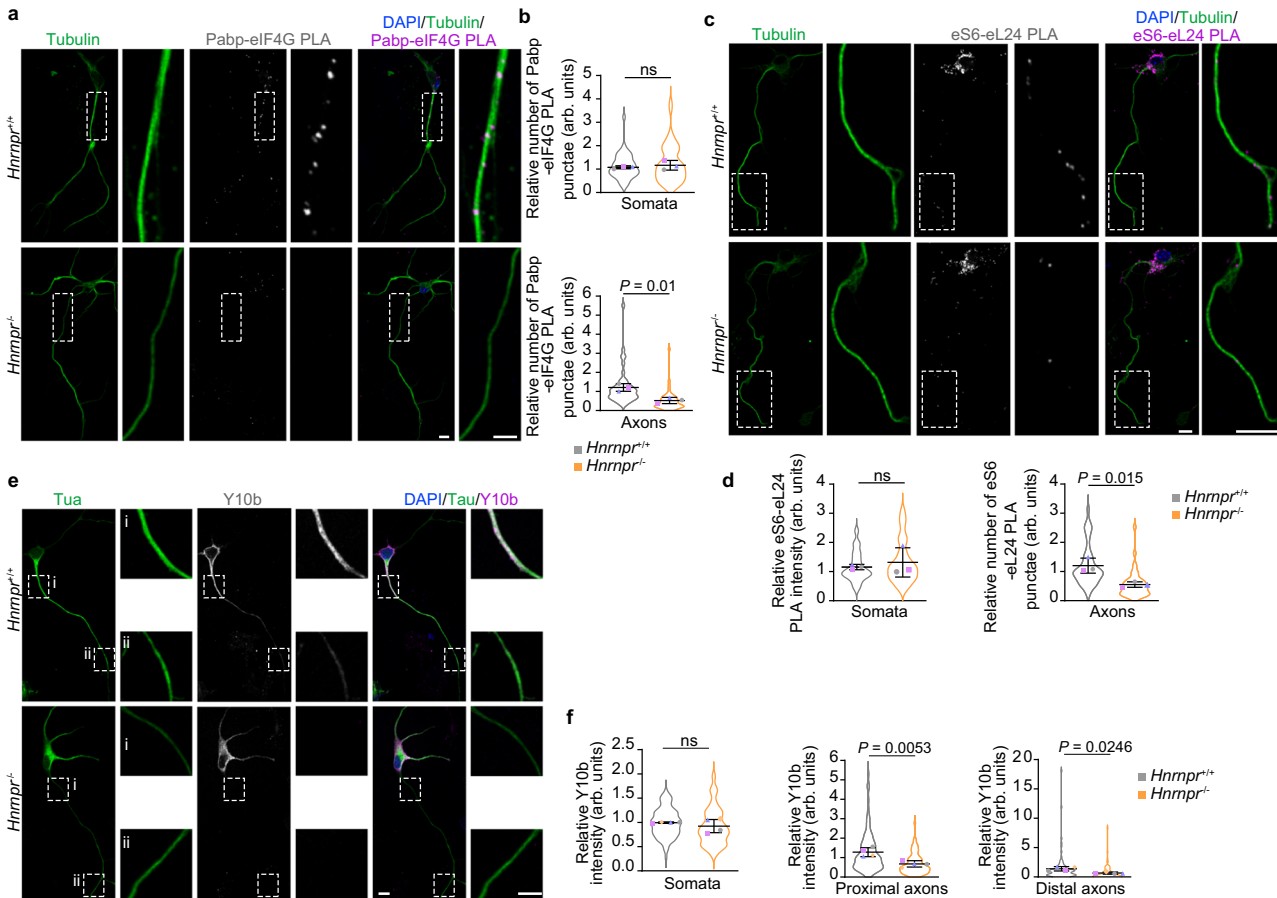

**Fig. 6 | Translation initiation is impaired in axons of *Hnrnpr*⁻/⁻ motoneurons.**
**a** Representative images of Pabp-eIF4G PLA signal in cultured *Hnrnpr*⁻/⁻ and +/+ motoneurons at DIV 6 using anti-Pabp and anti-eIF4G antibodies. Scale bars, 10 μm and 5 μm (magnified areas). **b**, SuperPlots of the relative number of Pabp-eIF4G PLA punctae in somata and 50 μm of axons of cultured *Hnrnpr*⁻/⁻ and +/+ motoneurons. Statistical analysis was performed using an unpaired two-tailed Student's *t* test. Data are mean ± s.d. of three biological replicates. *Hnrnpr*⁺/⁺, *n* = 44; *Hnrnpr*⁻/⁻, *n* = 51 motoneurons. **c** Representative images of eS6-eL24 PLA signal in cultured *Hnrnpr*⁻/⁻ and +/+ motoneurons at DIV 6 using anti-eS6 and anti-eL24 antibodies. Scale bars, 10 μm and 5 μm (magnified areas). **d** SuperPlots of the relative eS6-eL24 PLA signal

intensity in somata and number of punctae in 50 μm of axons of cultured *Hnrnpr*⁻/⁻ and +/+ motoneurons. Statistical analysis was performed using an unpaired two-tailed Student's *t* test. Data are mean ± s.d. of three biological replicates. *Hnrnpr*⁺/⁺, *n* = 44; *Hnrnpr*⁻/⁻, *n* = 41 motoneurons. **e** Representative images of Y10b immunostaining in cultured *Hnrnpr*⁻/⁻ and +/+ motoneurons at DIV 6. Scale bars, 10 μm and 5 μm (magnified areas). **f** SuperPlots of the relative Y10b immunosignal intensity in somata and proximal and distal axonal regions of cultured *Hnrnpr*⁻/⁻ and +/+ motoneurons. Statistical analysis was performed using an unpaired two-tailed Student's *t* test. Data are mean ± s.d. of three biological replicates. *Hnrnpr*⁺/⁺, *n* = 97; *Hnrnpr*⁻/⁻, *n* = 78 motoneurons. Source data are provided as a Source Data file.

strength and RotaRod analysis (Fig. 9b–d). This motor dysfunction persisted but did not significantly exacerbate over time, indicating that hnRNP R is essential for sustaining appropriate motor behavior.

Next, we evaluated whether the observed motor defects are accompanied by abnormalities of neuromuscular junctions (NMJs). For this purpose, we examined NMJs in the tibialis anterior (TA) muscle from *Hnrnpr*⁻/⁻ and +/+ mice aged 16 months. TA muscle is innervated by fast-fatigable motoneurons that are more susceptible to degeneration in ALS[29]. NMJs were visualized by fluorescently labeled α-bungarotoxin (αBTX) to visualize postsynaptic acetylcholine-receptors, an antibody against Synaptophysin 1 to label presynaptic nerve terminals and an antibody against Neurofilament heavy chain to label axons (Fig. 9e). The apposition of αBTX and Synaptophysin 1 signals was used to score NMJ innervation, revealing that denervated NMJs were significantly more frequent in *Hnrnpr*⁻/⁻ compared to +/+ mice (Fig. 9f). In contrast, there was no significant difference in motoneuron numbers in the lumbar region of spinal cords of *Hnrnpr*⁻/⁻ and +/+ mice (Supplementary Fig. 6a–c). Thus, hnRNP R deficiency leads to NMJ defects without overt motoneuron loss.

## Discussion

Local protein synthesis allows axons to respond rapidly to extracellular guidance cues during development, and contributes to axon regeneration following nerve injury[30–33]. High-throughput sequencing studies have revealed many mRNAs that localize in axons and mechanisms are beginning to be understood that underlie their axonal delivery[34]. RBPs function as trans-acting factors that package mRNAs into larger mRNPs that are transported as cargo by kinesins[35,36]. Conspicuously, while components of the translational machinery are often part of such granules, mRNAs are maintained in a translationally repressed state during transport, allowing local activation of protein synthesis at target sites in growth cones or at synapses[37,38]. Except for individual transcripts such as the mRNA encoding β-actin[39], the mechanisms regulating axonal translation are largely unknown. RBPs have previously been shown to regulate translation by facilitating the formation of pre-initiation complexes, or by strengthening or repressing interactions between the mRNA and the ribosome[40]. For example, we recently showed that the neuronal RBP Ptbp2 promotes axonal translation of *Hnrnpr* mRNA by facilitating its association with translating ribosomes[17]. Translation initiation involves the formation of an

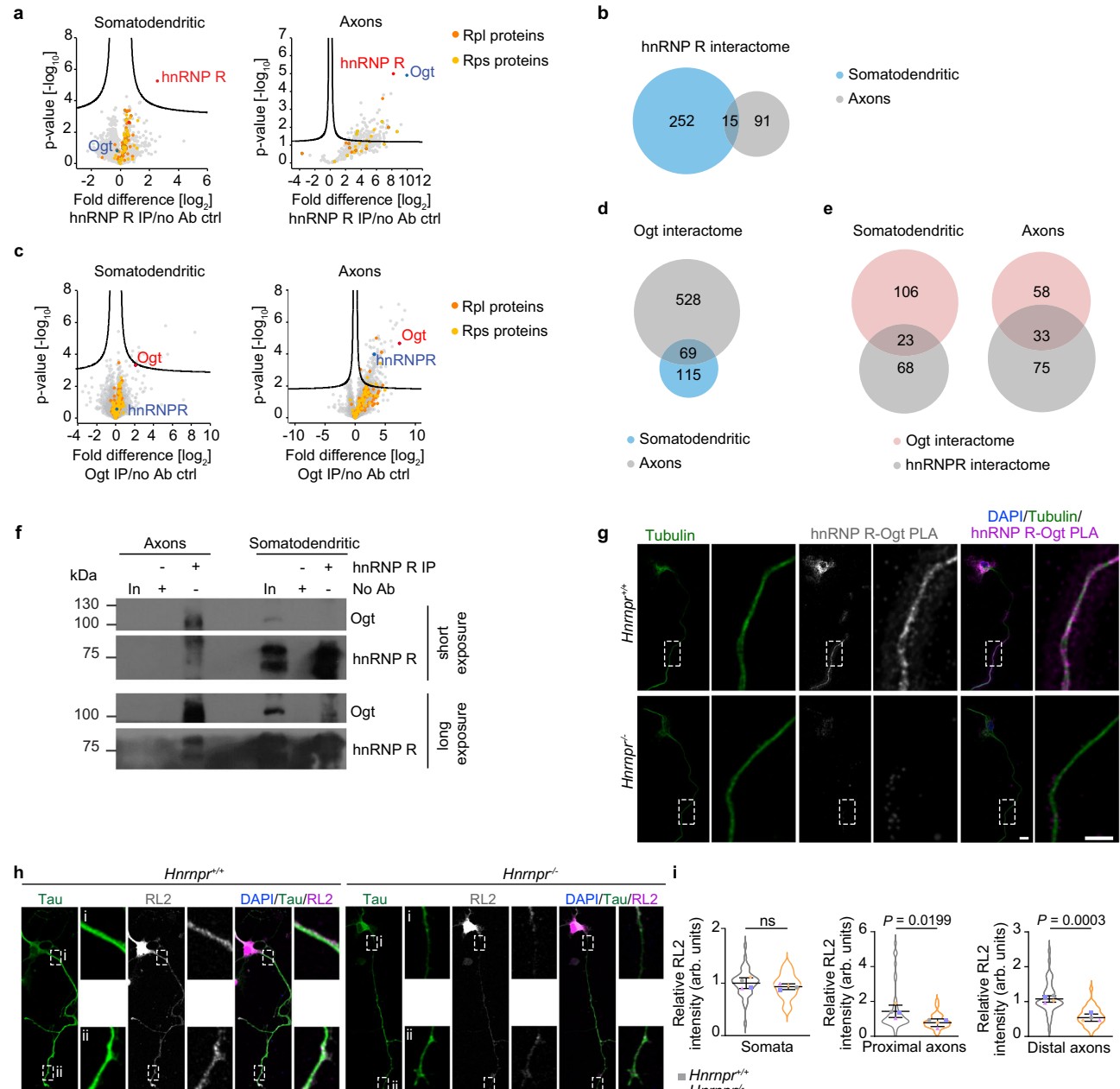

**Fig. 7 | hnRNP R associates with Ogt in axons and is required for O-GlcNAcylation of axonal proteins. a** Volcano plots showing enrichment of proteins identified by mass spectrometry after immunoprecipitation with a C-terminal-specific anti-hnRNP R antibody relative to no antibody (no Ab) control purifications from four biological replicates of somatodendritic and axonal lysate of motoneurons cultured in microfluidic chambers. Proteins of the large and small ribosomal subunit are shown in orange and yellow, respectively. **b** Venn diagrams showing overlap of proteins significantly ($P < 0.05$) associated with hnRNP R in the somatodendritic and axonal compartments of motoneurons. Only proteins detected in both datasets were considered for the analysis. **c** Volcano plots of protein enrichment detected by mass spectrometry after Ogt immunoprecipitation from four biological replicates of somatodendritic and axonal lysates of moto-neurons cultured in microfluidic chambers. Proteins of the large and small ribo-somal subunit are shown in orange and yellow, respectively. **d** Venn diagrams showing overlap of proteins significantly ($P < 0.05$) associated with Ogt in the somatodendritic and axonal compartments of motoneurons. Only proteins

detected in both datasets were considered for the analysis. **e** Venn diagrams showing the overlap of Ogt and hnRNP R interactome datasets from somatoden-dritic and axonal compartments of motoneurons. Only proteins detected in both datasets were considered for the analysis. **f** Immunoblot analysis of Ogt co-immunoprecipitated by anti-hnRNP R from somatodendritic and axonal lysates of motoneurons. The immunoblots are representative of two biological replicates. **g** Representative images of hnRNP R-Ogt PLA signal in cultured *Hnrnpr⁻/⁻* and +/+ motoneurons at DIV 6. Scale bars, 10 μm and 5 μm (magnified areas). The images are representative of three biological replicates. **h** Representative images of O-GlcNAc immunostaining using the RL2 antibody in cultured *Hnrnpr⁻/⁻* and +/+ motoneurons at DIV 6. Scale bars, 10 μm and 5 μm (magnified areas). **i** SuperPlots of the relative RL2 immunosignal intensity in somata and axons of cultured *Hnrnpr⁻/⁻* and +/+ motoneurons. Statistical analysis was performed using an unpaired two-tailed Student's *t* test. Data are mean ± s.d. of three biological replicates. *Hnrnpr⁺/⁺*, *n* = 42; *Hnrnpr⁻/⁻*, *n* = 39 motoneurons. Source data are provided as a Source Data file.

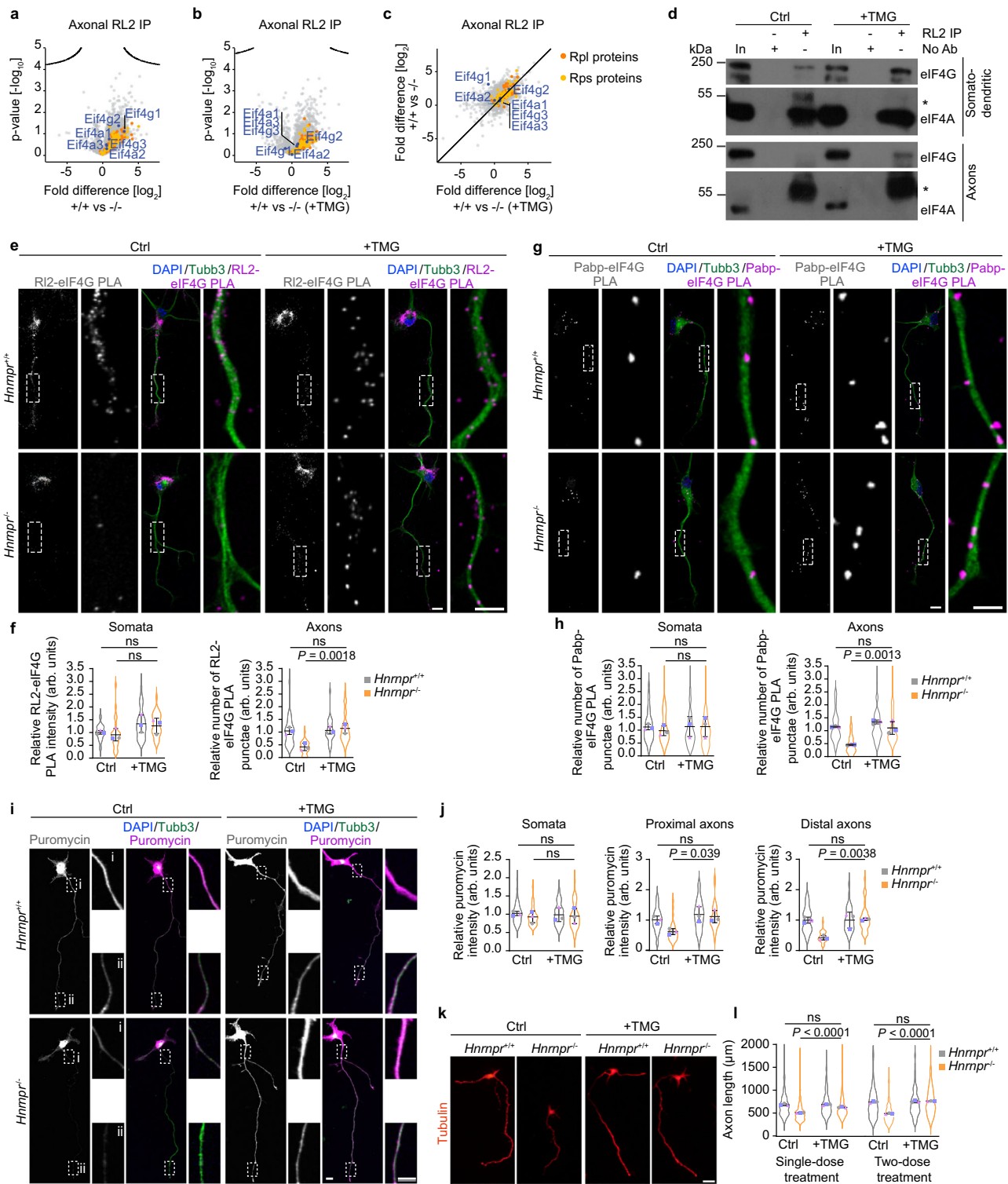

mRNA "closed loop" through interaction of eIF4G with PABP, and this pseudo-circularization of mRNA can be modulated by RBPs at different stages[41]. However, mechanisms regulating translation initiation have largely been studied in whole cells and very little is known about such processes in subcellular regions of neurons.

Here, we uncovered a role for hnRNP R as a regulator of axonal translation initiation in motoneurons. We deployed a range of methods to show that hnRNP R regulates axonal translation in vitro and in vivo. We observed that depletion of hnRNP R in motoneurons leads to markedly decreased protein synthesis exclusively in axons but not

the somatodendritic compartment in cultured motoneurons of *Hnrnpr⁻/⁻* mice.

To shed light on the mechanisms underlying hnRNP R's role in axonal translation regulation, we performed mass spectrometry on axonal lysates derived from *Hnrnpr⁻/⁻* and +/+ mouse motoneurons, detecting widespread changes in protein abundance in hnRNP R-deficient axons. Additionally, our RNA-seq data from somatodendritic and axonal compartments of hnRNP R knockout motoneurons revealed that these axonal proteome alterations are not simply due to changes in the axonal abundance of the encoding transcripts.

**Fig. 8 | hnRNP R promotes Ogt-mediated O-GlcNAcylation of eIF4G. a** Volcano plots showing differential enrichment of O-GlcNAcylated proteins obtained by RL2 immunoprecipitation from three biological replicates of axonal lysates of *Hnrnpr*$^{-/-}$ compared to +/+ motoneurons cultured in microfluidic chambers. **b** Volcano plots showing differential enrichment of O-GlcNAcylated proteins from three biological replicates of axonal lysates of thiamet G (TMG)-treated *Hnrnpr*$^{-/-}$ motoneurons compared to +/+ motoneurons cultured in microfluidic chambers. **c** Scatter plot of fold differences from (**a**) and (**b**). **d** Immunoblot analysis of eIF4G and eIF4A co-immunoprecipitated by RL2 antibody from somatodendritic and axonal lysates of motoneurons treated with DMSO (Ctrl) or TMG. Asterisks mark antibody heavy chain. The immunoblots are representative of three biological replicates. **e** Representative images of RL2-eIF4G PLA signal in cultured DIV 6 *Hnrnpr*$^{-/-}$ and +/+ motoneurons treated with DMSO (Ctrl) or TMG. Scale bars, 10 μm and 5 μm (magnified areas). **f** SuperPlots of the relative RL2-eIF4G PLA signal intensity in somata and number of punctae in 50 μm of axons of cultured DIV 6 *Hnrnpr*$^{-/-}$ and +/+ motoneurons treated with DMSO (Ctrl) or TMG. Ctrl: *Hnrnpr*$^{+/+}$, *n* = 29; *Hnrnpr*$^{-/-}$, *n* = 31 motoneurons; TMG: *Hnrnpr*$^{+/+}$, *n* = 35; *Hnrnpr*$^{-/-}$, *n* = 37 motoneurons. **g** Representative images of Pabp-eIF4G PLA signal in cultured DIV 6 *Hnrnpr*$^{-/-}$ and +/+ motoneurons treated with DMSO (Ctrl) or TMG. Scale bars, 10 μm and 5 μm

(magnified areas). **h** SuperPlots of the relative number of Pabp-eIF4G PLA punctae in somata and 50 μm of axons of cultured DIV 6 *Hnrnpr*$^{-/-}$ and +/+ motoneurons treated with DMSO (Ctrl) or TMG. Ctrl: *Hnrnpr*$^{+/+}$, *n* = 48; *Hnrnpr*$^{-/-}$, *n* = 54 motoneurons; TMG: *Hnrnpr*$^{+/+}$, *n* = 52; *Hnrnpr*$^{-/-}$, *n* = 50 motoneurons. **i** Representative images of puromycin immunostaining in cultured DIV 6 *Hnrnpr*$^{-/-}$ and +/+ motoneurons treated with DMSO (Ctrl) or TMG. Scale bars, 10 μm and 5 μm (magnified areas). **j** SuperPlots of the relative puromycin immunosignal intensity in somata and axons of cultured DIV 6 *Hnrnpr*$^{-/-}$ and +/+ motoneurons treated with DMSO (Ctrl) or TMG. Ctrl: *Hnrnpr*$^{+/+}$, *n* = 36; *Hnrnpr*$^{-/-}$, *n* = 33 motoneurons. TMG: *Hnrnpr*$^{+/+}$, *n* = 33; *Hnrnpr*$^{-/-}$, *n* = 35 motoneurons. **k** Morphology of cultured motoneurons from *Hnrnpr*$^{-/-}$ and +/+ mice treated with DMSO (Ctrl) or TMG immunostained with an antibody against tau. Scale bar, 50 μm. **l** SuperPlots of axon lengths of cultured *Hnrnpr*$^{-/-}$ and +/+ motoneurons treated with a single dose for 24 h or two doses for 48 h of DMSO (Ctrl) or TMG. Single dose: Ctrl: *Hnrnpr*$^{+/+}$, *n* = 208; *Hnrnpr*$^{-/-}$, *n* = 202 motoneurons. TMG: *Hnrnpr*$^{+/+}$, *n* = 234; *Hnrnpr*$^{-/-}$, *n* = 172 motoneurons. Two doses: Ctrl: *Hnrnpr*$^{+/+}$, *n* = 210; *Hnrnpr*$^{-/-}$, *n* = 217 motoneurons. TMG: *Hnrnpr*$^{+/+}$, *n* = 185; *Hnrnpr*$^{-/-}$, *n* = 159 motoneurons. For **f**, **h**, **j**, **l** statistical analysis was performed using a two-way ANOVA with Tukey's multiple comparisons test. Data are mean ± s.d. of three biological replicates. Source data are provided as a Source Data file.

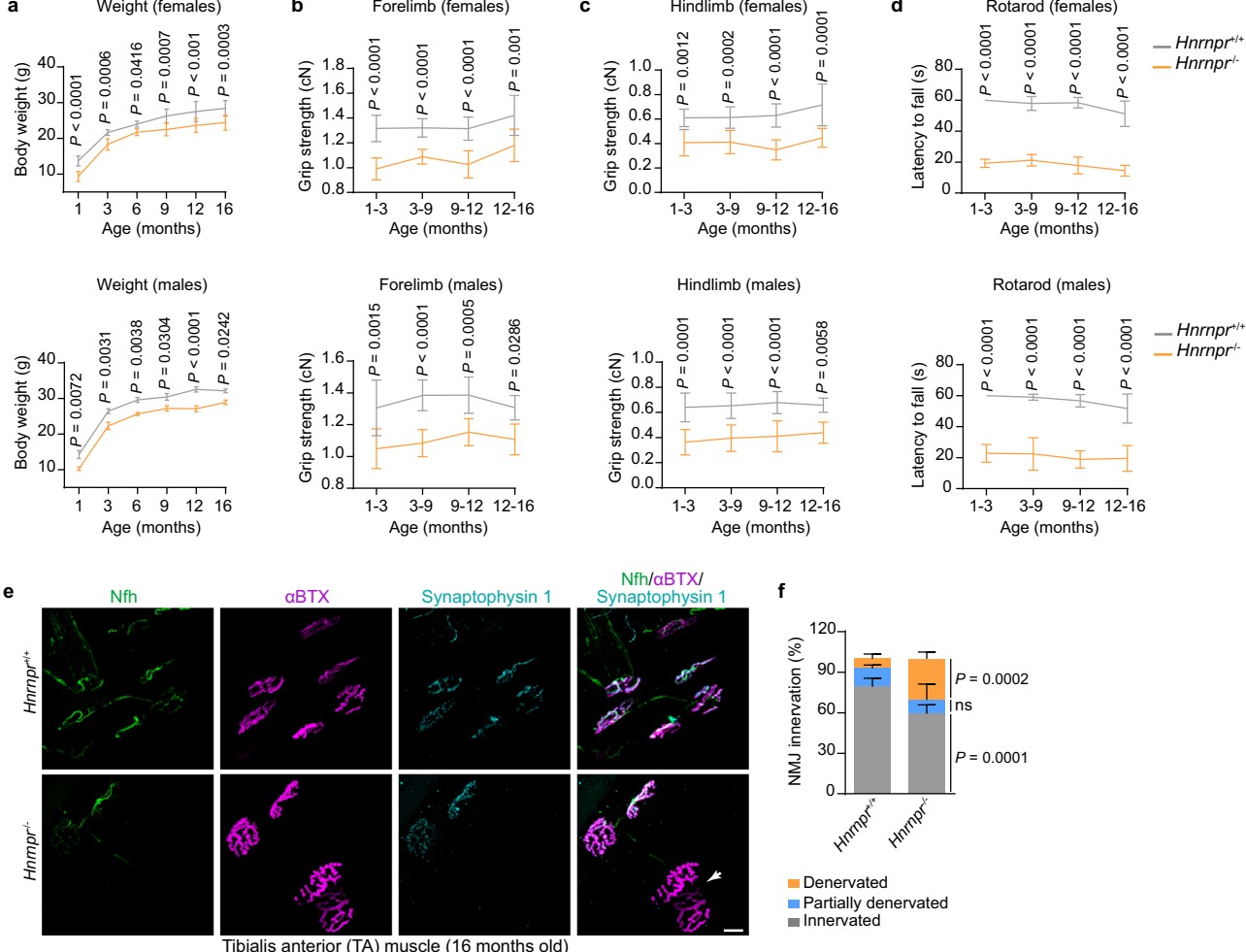

**Fig. 9 | hnRNP R mutant mice display motor deficits. a–d** Body weight (**a**), grip strength of forelimbs (**b**) and hindlimbs (**c**) and RotaRod performance (**d**) of female and male *Hnrnpr*$^{-/-}$ and +/+ mice. Statistical analysis was performed using a two-way ANOVA with Sidak's multiple comparisons test. Data are mean ± s.d of *n* = 9/9 mice per *Hnrnpr*$^{+/+}$/*Hnrnpr*$^{-/-}$ for females and *n* = 9/7 mice per *Hnrnpr*$^{+/+}$/*Hnrnpr*$^{-/-}$ for males. **e**, Tibialis anterior (TA) muscle stained with α-bungarotoxin (αBTX) and

antibodies against Synaptophysin 1 and Neurofilament heavy chain (Nfh). Scale bar, 20 μm. **f**, Quantification of neuromuscular junction (NMJ) denervation in *Hnrnpr*$^{-/-}$ and +/+ mice. The arrow denotes a denervated NMJ. Statistical analysis was performed using a two-way ANOVA with Sidak's multiple comparisons test. Data are mean ± s.d. of four biological replicates. Source data are provided as a Source Data file.

Therefore, while hnRNP R has functions in axonal mRNA translocation, it exerts functionally separate roles in local translation. Among proteins dysregulated in axons of *Hnrnpr*[−/−] motoneurons, we observed many components of the translational machinery, including ribosomal proteins and translation factors. For example, we observed reduced levels of eIF3c and eS3, both of which have been implicated in translation initiation. Knockdown of *eIF3c* reduces the efficiency of mRNA recruitment to 43S pre-initiation complexes[42]. eS3 is involved in intra-ribosomal rearrangements of the 40S subunit following binding of eIF1 and eIF1A, thereby opening the ribosomal mRNA binding channel[43]. Thus, reduction in eIF3c, eS3, and possibly other translation components might contribute to the lowered protein synthesis in hnRNP R-deficient axons. Conspicuously, we also observed an upregulation of several ribosomal proteins and translation factors in axons of hnRNP R-deficient motoneurons. These alterations might represent compensatory mechanisms to counteract the reduced axonal translation induced by hnRNP R loss.

In agreement with our previous hnRNP R interactome analysis[18], we observed that hnRNP R binds to ribosomes as well as to the translation initiation factor eIF4G. While its association with ribosomes was RNA-dependent, hnRNP R binding to eIF4G was mediated by protein contacts. Since hnRNP R binds to transcripts at their 3′UTR[16], this finding puts forward a model according to which axonal hnRNP R bound to 3′UTRs stimulates translation initiation by facilitating mRNA circularization via interaction with eIF4G, thereby promoting its association with Pabp and stimulating 48S pre-initiation complex formation. We obtained several lines of evidence to support this notion. First, using polysome fractionation of axonal lysate, we discovered that hnRNP R associates with the translational machinery at the level of ribosomal subunits and monosomes in axons of motoneurons. Second, we observed that loss of hnRNP R reduces the interaction between eIF4G and Pabp in axons but not motoneuron cell bodies. Finally, we performed proteomic analysis to determine the interactome of axonal hnRNP R. In addition to ribosomal proteins, we identified Ogt as a major interactor of hnRNP R in axons but not cell bodies of motoneurons. Ogt-mediated O-GlcNAcylation is prevalent in the brain, and many neuronal and synaptic proteins are modified by O-GlcNAc addition[44]. We observed reduced O-GlcNAc levels in axons of *Hnrnpr*[−/−] motoneurons that could be restored by treatment with the Oga inhibitor thiamet-G. Importantly, O-GlcNAcylation of eIF4G, which promotes its binding to PABP and, thus, mRNA circularization, was reduced in hnRNP R-deficient axons. Thiamet-G treatment could rescue axonal O-GlcNAcylation of eIF4G and stimulate eIF4G binding to Pabp, accompanied by restored axonal protein synthesis, in hnRNP R-deficient motoneurons. This suggests that hnRNP R is required for Ogt-mediated O-GlcNAcylation of eIF4G, which increases its binding to Pabp. Interestingly, hnRNP R itself has recently been shown to be locally synthesized from axonal *Hnrnpr* transcripts in a manner dependent on the RBP Ptbp2, indicating cross-regulation of axonal RBPs to fine-tune axonal protein synthesis[17].

Besides ribosomal proteins and translation factors, we detected reduced levels of proteins with functions in synapse formation and cytoskeleton assembly in axons of *Hnrnpr*[−/−] motoneurons, in agreement with a role of hnRNP R in axon growth. We identified Macf1 as the most strongly downregulated proteins in axons of hnRNP R-deficient motoneurons. Macf1 is a microtubule-associated protein that stabilizes NMJs and is also involved in axonal vesicular trafficking and axon growth[45–47]. MACF1 dysregulation or mutations have been linked to neurological diseases including Parkinson's disease (PD), autism spectrum disorder (ASD), bipolar disorder and schizophrenia[48–51]. Stmn2 is another microtubule-regulating protein that was reduced in axons of hnRNP R-depleted motoneurons (Supplementary dataset 2). The *STMN2* transcript is tightly regulated by TDP-43 and loss of STMN2 function occurs downstream of pathological TDP-43 alterations in ALS, leading to loss of NMJ

innervation[52,53]. Considering that hnRNP R itself is an interactor of TDP-43[54], and that loss of hnRNP R leads to reduced Stmn2 levels in axons, it is thus possible that hnRNP R dysregulation contributes to the axonal pathomechanisms underlying ALS.

In summary, our study provides evidence that hnRNP R is associated with ribosomes at the translation initiation stage in axons of motoneurons and that perturbation of this function leads to axonal translation defects resulting in axon growth impairments in vitro and NMJ dysfunction in vivo (Fig. 10). *Hnrnpr*[−/−] mice have reduced motor function and exhibit NMJ denervation in the TA muscle innervated by fast-fatigable motoneurons that have been reported to be more sensitive to degeneration in ALS[29]. Importantly, we did not observe loss of motoneurons in spinal cords of *Hnrnpr*[−/−] mice, which is similar to *Stmn2* knockout mice showing NMJ defects without motoneuron degeneration[52]. Our data thus show that axonal translation dysfunction affects motor behavior and NMJ integrity but does not necessarily reduce motoneuron survival. Research over the past years has revealed that many proteins involved in motoneuron diseases function in local translation. For example, the SMN protein, which is deficient in spinal muscular atrophy, associates with ribosomes and regulates translation of a specific subset of mRNAs relevant to the pathogenesis of SMA[55]. Additionally, TDP-43 regulates the translation of several transcripts and, in ALS motoneurons, defects in axonal translation have been observed[11,56,57]. Similarly, FUS-ALS mutants can impair axonal protein translation[58,59]. Given the fact that NMJ pathology is considered an early event in ALS pathogenesis[60] and that hnRNP R plays a role in the pathogenesis of neurodegenerative diseases such as frontotemporal dementia and ALS[61–63], our study highlights the importance of local translation for maintaining the integrity of the neuromuscular system and points towards the possibility that hnRNP R dysfunction might contribute to the pathology underlying certain forms of neurodegenerative disorders.

## Methods

### Ethics statement

All of the experimental procedures in this study involving mice were performed according to the regulations on animal protection of the German federal law and the Association for Assessment and Accreditation of Laboratory Animal Care, in agreement with and under the control of the local veterinary authority. Mice were housed in the animal facility of the Institute of Clinical Neurobiology at the University Hospital of Wuerzburg. Mice were maintained on a 12 h/12 h day/night cycle under controlled conditions at 20-22 °C and 55-65% humidity with food and water in abundant supply.

### Generation of *Hnrnpr* knockout mice

*Hnrnpr* knockout mice were generated by Cyagen. Briefly, two gRNAs together with Cas9 mRNA were co-injected into fertilized mouse eggs to delete exons 1-5 of the *Hnrnpr* gene. The target sequences were: gRNA1: 5′-ATATGCATAGTCTACTGCCCTGG−3′; gRNA2: 5′-CCCA-GACTACCTTGACGATGGGG−3′ (PAM sequences are underlined). The deletion was verified by sequencing. PCR genotyping was performed using the following primers: F[C]: 5′-TACCTTTGATGAAGGACTGGA-GAGTTAGC-3′; R[Wt/He]: 5′-TTCTTCCTCTGACCTTCGTCTGCAC-3′; R[KO]: 5′-ACTCTCCTTCATGGGATTCCCTCC-3′. Mice were backcrossed for at least four generations onto C57Bl/6 background.

### NSC-34 cell culture

NSC-34 cells (Cedarlane, cat. no. CLU140) were cultured in high glucose Dulbecco's modified Eagle's Medium (DMEM; Gibco) supplemented with 10% fetal calf serum (Gibco), 2 mM GlutaMAX (Gibco) and 1% penicillin-streptomycin (Gibco) in a humidified incubator at 37 °C with 5% $CO_2$. Cells were splitted when they were 70% confluent. Cells were tested negative for mycoplasma contamination.

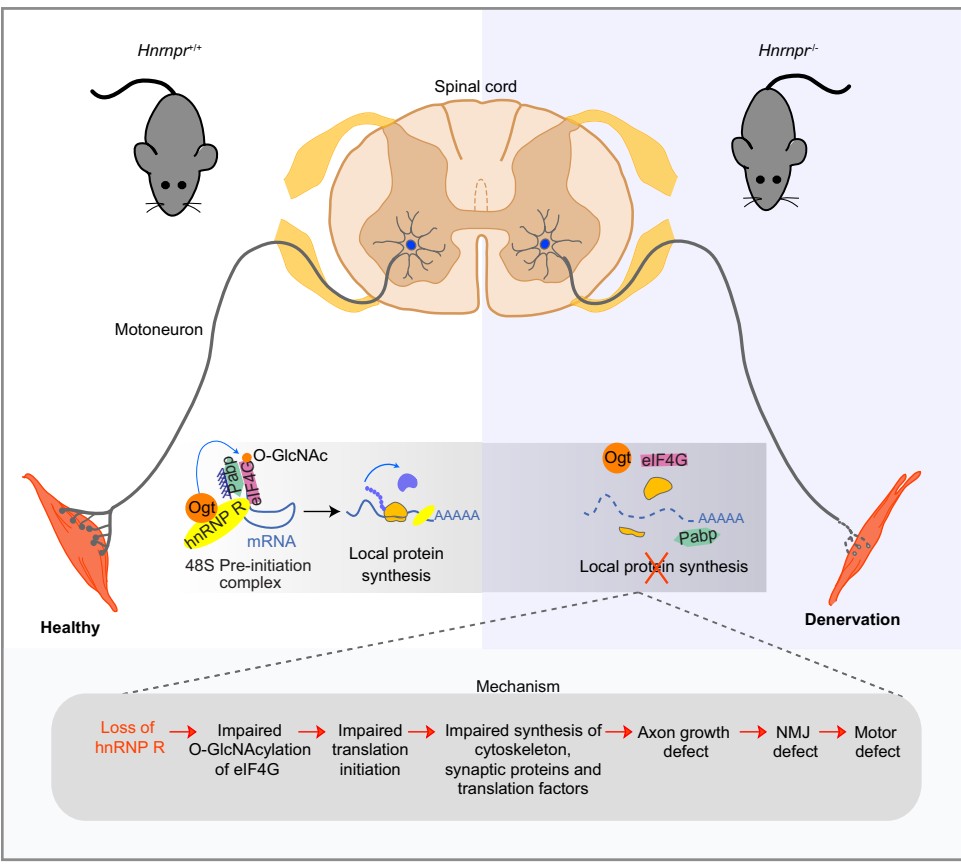

**Fig. 10 |** Schematic summary of hnRNP R functions in axon of motoneurons.

## Primary embryonic mouse motoneuron culture

Isolation and enrichment of primary mouse motoneurons were performed as previously described[64]. Briefly, lumbar spinal cords were isolated from E13 mouse embryos, and motoneurons were enriched by panning using a p75[NTR] antibody (clone MLR2, Biosensis). Cells were plated on coverslips or culture dishes coated with poly-DL-ornithine hydrobromide (PORN) (Sigma-Aldrich, P8638) and laminin-111 (Thermo Fisher Scientific, 23017-015). Motoneurons were maintained at 37 °C, 5% $CO_2$ in Neurobasal medium (Gibco) supplemented with 2% B27 (Gibco), 2% heat-inactivated horse serum (Gibco), 500 µM Gluta-MAX (Gibco) and 5 ng/ml of brain-derived neurotrophic factor (BDNF). The medium was replaced one day after plating and then every second day. For analysis of motoneuron survival, $Hnrnpr^{-/-}$ and +/+ motoneurons were cultured on laminin-111-coated glass coverslips for 6 DIV. Number of surviving neurons that were identified by morphological appearance of the cell body and intact neurites were quantified manually in defined areas on each coverslip on DIV 2, 4, and 6.

For compartmentalization of motoneurons, two different microfluidic chambers were used: Xona Microfluidics SND150 as described previously[9] and Xona Microfluidics IND150, which were pre-coated with 0.1% poly-L-lysine hydrobromide (Sigma-Alrich, P2636) and laminin-111. 600,000 motoneurons were directly plated in the somatodendritic compartment and CNTF (5 ng/ml) was applied to both the somatodendritic and the axonal compartment. Axon growth was stimulated by adding 20 ng/ml BDNF only to the axonal compartment. Culture medium was exchanged on day 1 and then every second day.

## Lentivirus transduction

Lentivirus particles were packaged in HEK293TN cells (System Biosciences, cat. no. LV900A-1) cells with pCMV-pRRE, pCMV-pRSV, and pCMV-pMD2G as previously described[65]. Transduction was done by incubation of motoneurons with lentiviruses in a total volume of 50 µl for 10 min at room temperature before plating at DIV 0.

## RNA extraction

Total RNA was purified from primary mouse motoneurons and mouse brain tissue using the NucleoSpin RNA kit (Macherey-Nagel, 740955.50). For brain tissue, samples were collected in microcentrifuge tubes and immediately snap-frozen by immersion in liquid nitrogen. Samples were kept at −80 °C until processing. For motoneurons cultured in microfluidic chambers, RNA was purified from the somatodendritic and axonal compartment using the PicoPure RNA Isolation Kit (Thermo Fisher Scientific, KIT0204) with 10 µl elution volume. 1 µl of somatodendritic RNA and 10 µl of axonal RNA were used for reverse transcription.

## cDNA synthesis and qPCR analysis

cDNA synthesis was conducted using the First Strand cDNA Synthesis Kit (Thermo Fisher Scientific, K1612) using random hexamers. Reverse transcription reactions were diluted 1:5 in RNase-free water. qPCR reactions were performed with the Luminaris HiGreen qPCR Master Mix (Thermo Fisher Scientific, K0994) and run on a LightCycler 96 (Roche). qPCR primers were designed using the online Primer3Plus design tool and the sequences are listed in Supplementary dataset 9. Relative expression was calculated using the ΔΔCt method.

## Co-immunoprecipitation

For NSC-34 cells, cells were plated in 10 cm dishes until 80% confluence. Cells were washed once with PBS and lysed in lysis buffer (10 mM HEPES pH 7.0, 100 mM KCl, 5 mM MgCl2, 0.5% NP-40) on ice for 20 min and cleared via centrifugation at 20,000 × $g$ for 10 min at 4 °C followed by incubation with 1 U of Benzonase (Santa Cruz

Biotechnology) at 4 °C for 30 min on a rotator to allow complete digestion of nucleic acids. Magnetic Dynabeads Protein A or G were bound to antibody or nonspecific IgG by rotating for 1 h at room temperature. 300 µl lysate was added to the antibody-bound beads and rotated overnight at 4 °C. Beads were washed twice with 500 µl wash buffer (20 mM Tris-HCl, 150 mM KCl, 2 mM MgCl₂, 0.1% NP-40) and proteins were eluted in 1× Laemmli buffer. Proteins were size-separated by SDS-PAGE and analyzed by immunoblotting.

For primary mouse motoneurons, cells were grown on laminin-111-coated 6 cm dishes for 7 DIV and then processed as described above for NSC-34 cells. Antibodies used and their dilutions are listed in Supplementary dataset 11.

## Ribosome pulldown

For NSC-34 cells, cells were grown in 10 cm dishes until 80% confluence. The medium was supplemented with 100 µg/ml cycloheximide and cells were incubated for 10 min in a humidified incubator at 37 °C. Cells were washed twice with ice-cold Hanks' Balanced Salt Solution (HBSS) containing 100 µg/ml cycloheximide and lysed in 1 ml lysis buffer (20 mM Tris-HCl, 150 mM KCl, 2 mM MgCl₂, 0.1% NP-40, 100 µg/ml cycloheximide). The lysate was incubated for 20 min on ice and cleared by centrifugation at $20,000 \times g$, 4 °C for 10 min followed by incubation with RNase I (7.5 U per unit absorbance at 260 nm of lysate) at room temperature for 45 min. 10 µl of magnetic Dynabeads Protein G were bound to 1 µg Y10b antibody (Santa Cruz Biotechnology, sc-33678) or 1 µg mouse IgG control (Santa Cruz Biotechnology, sc-2025) in lysis buffer containing 100 µg/ml yeast tRNA for 1 h at room temperature by rotation. 100 µl of lysate was used as an input sample, and 450 µl of lysate was incubated with the Y10b antibody- or IgG-bound beads overnight at 4 °C by rotation. Beads were washed twice with lysis buffer and eluted in 1× Laemmli buffer.

For primary mouse motoneurons, cells were cultured on laminin-111-coated 6 cm dishes for 7 DIV. The medium was supplemented with 100 µg/ml cycloheximide and cells were incubated for 10 min in a humidified incubator at 37 °C. Cells were washed twice with ice-cold HBSS containing 100 µg/ml cycloheximide and lysed in 700 µl of lysis buffer (20 mM Tris-HCl, 150 mM KCl, 2 mM MgCl₂, 0.1% NP-40, 100 µg/ml cycloheximide). The rest of the experiment was performed as for NSC-34 cells. Antibodies used and their dilutions are listed in Supplementary dataset 11.

For motoneurons cultured in microfluidic chambers (Xona Microfluidics, IND150), the medium in the compartments was supplemented with 100 µg/ml cycloheximide and the chamber was incubated for 10 min in a humidified incubator at 37 °C. Compartments were washed twice with ice-cold HBSS containing 100 µg/ml cycloheximide and lysed in 200 µl of lysis buffer (20 mM Tris-HCl, 150 mM KCl, 2 mM MgCl₂, 0.1% NP-40, 100 µg/ml cycloheximide). After 20 min incubation on ice, the lysates were cleared by centrifugation at $20,000 \times g$, 4 °C for 10 min. Magnetic Dynabeads Protein G were bound to 1 µg Y10b antibody or 1 µg IgG control for 1 h at room temperature by rotation. 150 µl lysate was added to the antibody-bound beads and rotated for 2 h at 4 °C. Total RNA from somatodendritic and axonal compartments was purified by adding 300 µl extraction buffer (Pico-Pure RNA Isolation Kit) and 300 µl absolute ethanol to both input and beads followed by RNA extraction according to the manufacturer's instructions. RNA was reverse-transcribed and evaluated by qPCR analysis. The primers are listed in Supplementary dataset 12.

## Sample preparation for mass spectrometry

Primary mouse motoneurons were cultured in microfluidic chambers (Xona Microfluidics, SND 150) for 7 DIV. Compartments were washed once with phosphate-buffered saline (PBS) and directly lysed in 100 µl lysis buffer (1% Na deoxycholate, 10 mM Tris Hcl, and protease inhibitor). Then, samples were snap-frozen in liquid N2 for mass spectrometry analysis.

For mass spectrometry analysis of immunoprecipitates, primary mouse motoneurons were cultured in microfluidic chambers (Xona Microfluidics, IND150). Compartments were washed once with phosphate-buffered saline (PBS) and directly lysed in 200 µl lysis buffer (10 mM HEPES pH 7.0, 100 mM KCl, 5 mM MgCl2, 0.5% NP-40 and Halt™ Protease and Phosphatase Inhibitor Cocktail). The lysates were incubated for 20 min on ice. Magnetic Dynabeads Protein A (Thermo Fisher Scientific) were bound to 1 µg hnRNP R antibody (Abcam, ab30930), Ogt antibody (Proteintech,11576-2-AP), and RL2 antibody (Abcam, ab2739) diluted in 300 µl wash buffer (20 mM Tris-HCl, 150 mM KCl, 2 mM MgCl2, 0.1% NP-40, and Halt™ Protease and Phosphatase Inhibitor Cocktail) for 1 h at room temperature by rotation. For RL2 immunoprecipitation, 1 µM TMG was also used in all buffers. Beads were washed twice with wash buffer, and 200 µl lysate was added to the beads and incubated overnight at 4 °C by rotation. Beads were washed twice with 500 µl wash buffer and once with 500 µl wash buffer without NP40. Beads were snap-frozen in liquid N2 and kept frozen until sample preparation for mass spectrometry analysis.

## Sample preparation for LC-MS/MS

For the analysis of the axonal and somatodendritic proteomes, four batches of $6 \times 10^5$ motoneurons cultured in microfluidic chambers were used. Each batch comprised somatodendritic and axonal compartments of $Hnrnpr^{-/-}$ and +/+ motoneurons, amounting to 16 samples in total. Cells were harvested in 200 µl of 1% (v/v) sodium deoxycholate in 10 mM Tris-HCl. Based on silver stain gels, the protein yield was estimated to be 12 µg in somatodendritic samples and 150 ng in axonal samples. To reduce disulfide bonds and alkylate the reduced thiols, 20 µl of a 50 mM Tris(2-carboxyethyl)phosphine (TCEP), 200 mM chloroacetamide (CAA), and 500 mM Tris-HCl pH8 solution was added. Proteins were denatured by boiling at 90 °C for 10 min. After cooling down to room temperature, 3 ng each of trypsin and LysC was added to axonal samples and, likewise, 250 ng each to somatodendritic samples and proteins were digested overnight (16 h). The digest was stopped by the addition of 800 µl 1% trifluoroacetic acid (TFA) in isopropanol. The samples were centrifuged at $16,000 \times g$ at 4 °C for 10 min to remove particles. The supernatant was used for peptide clean-up using a Styrene Divinylbenzene - Reversed Phase Sulfonate resin (SDB-RPS) in the STAGE-tip format[66]. Briefly, after peptide loading, the resin was washed twice with 200 µl 1% (v/v) TFA in isopropanol and twice with 200 µl 0.2% (v/v) TFA. Peptides were eluted with 60 µl 1% ammonia in 50% (v/v) acetonitrile. The eluate was evaporated to dryness in a vacuum concentrator. Peptides were re-suspended in 2% v/v acetonitrile, 0.5% v/v acetic acid, 0.1% v/v trifluoroacetic acid; axonal samples in 7 µl and somatodendritic samples in 10 µl. The peptide concentration in somatodendritic samples was measured with a Nanodrop 2000 spectrophotometer (Thermo Fisher Scientific) and the somatodendritic samples were diluted to a concentration of about 0.1 µg/µl. The peptides were stored at −20 °C until LC-MS/MS analysis.

For the analysis of axonal and somatodendritic interactomes, the beads were resuspended in 50 µl of 20% (v/v) acetonitrile 60 mM triethylammonium bicarbonate buffer at pH8.5 (TEAB; Sigma-Aldrich). Proteins were denatured at 90 °C for 10 min in a shaking incubator rotating at 1000 rpm, then the solution was cooled down to room temperature. The beads were dispersed by mild sonication with a Bioruptor sonicator (Diagenode) for 5 cycles of 15 s of sonication and 15 s with the tubes being submerged in a 4 °C water bath. For axonal samples, 3 ng and, for somatodendritic samples, 100 ng each of LysC and trypsin were added in a volume of 5 µl to digest the proteins. The digest was carried out overnight (17 h) at 37 °C in a shaking incubator rotating at 1000 rpm. The beads were separated by magnetic force and 30 µl of the digest were transferred to another tube. The peptide concentration in somatodendritic samples was estimated with a Nanodrop 2000 spectrophotometer, the concentration of axonal

samples was below the detection limit. The solutions of both sample types were diluted 1:10 with 0.1% formic acid (v/v) to reduce the acetonitrile concentration and additionally acidified with 20 μl of 50% (v/v) trifluoroacetic acid to ensure compatibility with Evotip loading. For final measurements, about 10 ng of peptide material was loaded onto Evotip Pure C18 tips (Evosep). The required volumes for 10 ng were estimated based on the observed total ion current (TIC) signals in initial experiments, which were guided by the Nanodrop measurements. The Evotips were activated by soaking in 1-propanol until the liquid level was 1-2 mm above the resin disc. All centrifugation steps were carried out at 700 × *g* for 1 min. The Evotips were washed with 50 μl of 99.9% (v/v) acetonitrile 0.1% formic acid, centrifuged to dryness and again activated with 1-propanol. Then the Evotips were washed twice with 50 μl 0.1% formic acid. To avoid drying of the disc during the manual peptide loading and inject the peptides into liquid, 70 μl of 0.1% formic acid were added to the Evotips and brought down by a brief centrifugation at 300 × *g*. Then, peptides were loaded and the Evotip centrifuged as before and washed with 50 μl of 0.1% formic acid. To keep the Evotips wet until measurement, 150 μl of 0.1% formic acid were added and briefly centrifuged at 300 × *g*. The loaded Evotips were stored at 4 °C until measurement.

## LC-MS/MS

For analysis of the somatodendritic and axonal proteome, peptides were separated on an EASY-nLC 1200 HPLC system (Thermo Fisher Scientific). For the chromatography, we used in-house packed columns (75 μm inner diameter, 50 cm length, and 1.9 μm C18 particles [Dr. Maisch GmbH]) and 120 min gradient of buffer A (0.5% formic acid) to buffer B (80% acetonitrile, 0.5% formic acid). For somatodendritic samples, 400 ng of peptide per sample were used for LC-MS/MS analysis. For axonal samples, where less protein was available, 6 of the 7 μl of sample volume were injected. The liquid chromatography gradient started at 3% B, increasing to 30% B in 95 min, further to 60% B in 5 min, to 95% B in 5 min, staying at 95% B for 5 min, decreasing to 5% B in 5 min and staying at 5% B for 5 min. The gradient was run at a flow rate of 300 nl/min and a temperature of 60 °C. A trapped ion mobility spectrometry (TIMS) quadrupole time of flight mass spectrometer (timsTOF Pro 2, Bruker) was directly coupled to the LC via an electrospray ion source (CaptiveSpray Ion Source, Bruker) using a 10 μm steel emitter (ZDV Sprayer 10, Bruker). The mass spectrometer was operated with the DIA-PASEF acquisition scheme with 16 diaPASEF scans ('standard scheme')[67]. Briefly, the TIMS was operated with an ion mobility range of 0.6 – 1.6 (1/K0 [V s/cm$^2$]) and an accumulation and ramp time of 100 ms. Precursors were isolated in 16 windows, overall ranging from 400 to 1200 m/z. The cycle time of that method is about 1.8 s.

Axonal and somatodendritic hnRNP R interactome peptides were separated on an Evosep One system via a 31 min Whisper gradient (Whisper40, Evosep) on an Aurora Elite CSI column (15 cm length, 75 μm inner diameter and 1.7 μm C18 particles, AUR3-15075C18-CSI, IonOpticks) at 50 °C. The mobile phases were 0.1% formic acid in LC-MS grade water (buffer A) and 0.1% formic acid in 99.9% acetonitrile (buffer B). The Evosep LC system was coupled to a trapped ion mobility spectrometry (TIMS) quadrupole time of flight SCP mass spectrometer (timsTOF SCP, Bruker) via a nano-electrospray ion source (CaptiveSpray Ion Source, Bruker). The MS was operated with a data-dependent acquisition PASEF (DDA-PASEF) scheme to isolate and fragment up to 10 precursors per topN acquisition cycle, covering a m/z rage of 100 to 1700 and an ion mobility of 1/K0 0.7 to 1.3 V s/cm$^2$. The accumulation and ramp time were set to 100 ms and the collision energy was a linear ramp from 20 eV at 1/K0 = 0.6 V s/cm$^2$ to 59 eV at 1/K0 = 1.6 V s/cm$^2$. Singly charged precursors were excluded by their position in the m/z-IM plane using a polygon shape, and precursor signals over an intensity threshold of 500 arbitrary units were picked for fragmentation. Precursors were actively excluded for 0.4 min when

reaching a target intensity of 20,000 arbitrary units and were isolated with a 2 Th window below *m/z* 700 and 3 Th above. The MS was operated in high-sensitivity mode.

Samples of the OGT and RL2 interactome were analyzed with the same chromatographic setup as those of the hnRNP IP interactome but with a more recent mass spectrometer, the Orbitrap Astral (Thermo Scientific). Additionally, a FAIMS device (High-Field Asymmetric Waveform Ion Mobility Spectrometry, Thermo Fisher) with a compensation voltage of −40V was used to filter ions entering the mass spectrometer. Due to its high sampling speed, the Orbitrap Astral was operated in data-independent acquisition mode with small 8 Th windows covering a precursor m/z range of 380 to 980. The maximum ion injection time was set to 18 ms. MS1 spectra were acquired in the Orbitrap analyzer at a resolution of 240,000 and MS2 spectra with the Astral analyzer.

## Proteomics data processing

To process DIA-PASEF MS raw files for the analysis of axonal and somatodendritic proteomes, we employed the Spectronaut software (Biognosys) version 15.3.210906, searching spectra against the UniProtKB mouse FASTA database (UP000000589) downloaded in September 2021 using canonical and isoform protein sequences amounting to 63,656 entries. The search was performed as a library-free ('direct-DIA') search. Default search parameters were utilized unless stated differently. Briefly, tryptic peptides ranging from 7 to 52 amino acids in length with a maximum of two miscleavages were identified. Carbamidomethylation on cysteine was fixed, acetylation of the protein N-terminus and oxidation of methionine were permitted as variable modifications. The FDR was set to 0.01 at peptide to spectrum match (PSM) level, peptide, and protein group level for the PULSAR search. For the DIA analysis, a precursor Q-value cutoff of 0.01 and posterior error probability (PEP) cutoff of 0.05 was used. On protein level, a sample ("Run") Q-value cutoff of 0.05 and a global ("Experiment") Q-value cutoff of 0.01 was used. Quantities were derived from MS2 level intensities, cross-run normalized with default settings, and MaxLFQ normalization algorithm was applied to protein quantities[68].

To process DDA-PASEF MS raw files for the analysis of axonal and somatodendritic interactomes of hnRNP R, we used MaxQuant version 2.3.1.0[69], searching against the above FASTA database for mouse. We set the data type to TIMS-DDA. Variable modifications and miscleavages were set as above, no fixed modifications were used. Identification by the 'Match-between runs' feature and quantification by the MaxLFQ algorithm was enabled. Otherwise, default parameters were used.

For the processing of DIA MS raw files for the analysis of OGT and RL2 interactomes, we used the DIA-NN software version 1.8.1., searching against the above FASTA database for mouse in a library-free fashion. The precursor mass range was adjusted to 380 – 980 and the variable modifications methionine oxidation, protein N-terminal methionine excision, and N-terminal acetylation were enabled. Matching between runs (MBR) and heuristic protein inference at the gene level were enabled, as well as the 'no shared spectra' option. MS1 and MS2 accuracy were set to 3 and 9ppm, according to the recommended parameters of an initial search with the option 'unrelated runs' ticked. Likewise, the scan window was set to 7ppm.

## Bioinformatic analysis of proteomic data

For the analysis of somatodendritic and axonal proteomes, the protein group abundances were exported from Spectronaut and loaded into the PERSEUS data analysis platform version 1.6.15.0 and log2-transformed[70]. For comparisons between genotypes within cellular compartments (axons/somatodendritic), the data were filtered for proteins with at least three valid values (experimental observations) in the four replicates of at least one group (*Hnrnpr*$^{-/-}$ or +/+) separately for axon and somatodendritic samples. This reduced the overall

dataset of 6960 protein groups to 2432 and 6897 in axon and soma samples, respectively. Missing values were imputed from a normal distribution using the default parameters (width = 0.3 and down-shift = 1.8 standard deviations of distribution of present intensities, separately in each sample) in PERSEUS. Then, batch correction was performed with the combat algorithm of the PyCombat package in Python, separately within the axon and soma data (https://epigenelabs.github.io/pyComBat/). For comparisons across the distinct cell compartment proteomes, the combined dataset was log2-transformed and filtered for at least six experimental observations in axon or soma samples (independent from *Hnrnpr*$^{-/-}$), reducing the dataset to 6893 protein groups. Imputation was performed above with the difference that the origin distribution (of present intensities) was modeled globally across all samples (mode 'total matrix') to avoid artefacts due to the strongly different number of identified proteins between axon and soma samples. Batch correction was not performed to avoid overcorrecting cell compartmental differences, however, batch correction was also not required due to the strong proteome differences between axon and soma compartments.

For the analysis of axonal and somatic hnRNP R interactomes, the protein group file outputted by MaxQuant was loaded into PERSEUS, and protein intensities were log2 transformed. Entries only identified by site, reverse (decoy) entries, and potential contaminants were removed. Separately for axon and soma, proteins were filtered for entries with at least two experimental observations out of four replicates in at least one condition (IP of wt, IP of KO, bead ctrl). This reduced the axon interactor dataset from 622 to 322 proteins and the soma interactor dataset from 1888 to 1845 proteins. Imputation was carried out as above. There was no apparent batch effect, hence no batch correction was employed. One bead control soma sample was an outlier as judged by principal component analysis and thus removed.

For the analysis of OGT and RL2 interactomes, the protein group matrix (pg.matrix) produced by DIA-NN was used for further analysis in PERSEUS. Protein intensities were log2-transformed, and the dataset was split for separate axon and soma interactome analysis as above. Much fewer proteins were detected in the bead control condition, which could lead to imputation artefacts. To prevent this, protein intensities were normalized by subtraction of the median within samples. Proteins were filtered for at least two observations across replicates of at least one experimental condition. Then, missing values values were imputed with the 'total matrix' option in PERSEUS.

Proteomic data was annotated with Gene Ontology (GO) terms for molecular function, cellular compartment, and biological processes and UniProt Keywords and further analyzed in PERSEUS. Volcano plots were generated with the built-in PERSEUS tool using *t*-test statistics combined with a permutation-based FDR of 5% and an s0-parameter of 0.1 for integrating the fold change effect size[71]. Proteins above the cutoff line have a q-value below 5%. Heatmaps were generated in PERSEUS for proteins identified in axon samples annotated as translation factors (GOMF translation factor, nucleic acid binding) or ribosomal proteins (UniProt Keywords). Fold changes (KO / WT in log2 space) for both axon and soma compartments and statistical significances for these comparisons are shown in the heatmap. The following proteins were used as custom cytoskeleton components listed in Supplementary dataset 4.

The 1D and 2D annotation enrichment analyses were performed with the built-in PERSEUS tool on the difference of group means (e.g. *Hnrnpr* KO - WT) of log2-transformed intensities[72]. In this enrichment analysis, positive enrichment scores correspond to higher intensities of in the first group (e.g. KO samples), whereas negative enrichment scores correspond to higher intensities in the second group (WT samples). For visualization, the enrichment score was plotted against the -log10-transformed Benjamini-Hochberg-adjusted p-values in case of the 1D annotation enrichment. For 2D annotation enrichment, terms were filtered for *p*-values of 5% or

lower and the enrichment scores were plotted for two fold-difference dimensions to be compared.

## RNA-seq

RNA was isolated from axonal and somatodendritic compartments of *Hnrnpr*$^{-/-}$ and +/+ motorneurons cultured in microfluidic chambers (Xona Microfluidics, IND150) using the PicoPure RNA isolation Kit. RNA from the somatodendritic compartment was evaluated using the 2100 Bioanalyzer with RNA 6000 Pico kit (Agilent Technologies) whereas RNA from axons was used directly due to the low amount. cDNA libraries suitable for sequencing were prepared with SMART-Seq® v4 Ultra® Low Input RNA Kit (Takara) according to the manufacturer's instructions (1/4 volume for the somatodendritic samples and full volume for the axon samples). The PCR amplification was performed using 10 PCR cycles for somatodendritic samples and 15 PCR cycles for axon samples. Libraries were quantified by Qubit™ dsDNA HS Assay Kit (3.0 Fluometer; Thermo Fisher Scientific) and quality was checked using the 2100 Bioanalyzer with High Sensitivity DNA kit. 0.5 ng of each library was subjected to a tagmentation-based protocol (Nextera XT, Illumina) using a quarter of the recommended reagent volumes. Libraries were quantified again by Qubit™ dsDNA HS Assay Kit and quality was checked using the 2100 Bioanalyzer with High Sensitivity DNA kit before pooling. Sequencing of pooled libraries spiked with 1% PhiX control library was performed in single-end mode with 100 nt read length on the NextSeq 2000 platform (Illumina) with a P2 Kit and $25 \times 10^6$ reads per library. Demultiplexed FASTQ files were generated with bcl2fastq2 v2.20.0.422 (Illumina).

To assure high sequence quality, Illumina reads were quality- and adapter-trimmed via Cutadapt[73] version 2.5 using a cutoff Phred score of 20 in NextSeq mode, and reads without any remaining bases were discarded (command line parameters: --nextseq-trim=20 -m 1 -a CTGTCTCTTATACACATCT). Processed reads were mapped to the mouse Mus musculus GRCm39 reference genome (RefSeq GCF_000001635.27_GRCm39) with splice junction-sensitive read mapper STAR version 2.7.2b using default parameters[74]. Read counts on exon level summarized for each gene were generated using featureCounts v1.6.4 from the Subread package[75]. Multi-mapping and multi-overlapping reads were counted strand-specific and reversely stranded with a fractional count for each alignment and overlapping feature (command line parameters: -s 2 -t exon -M -O --fraction).

The read abundance estimation and their differential expression were analyzed between axon and somatodendritic sample libraries using DESeq2[76] version 1.24.0. Read counts were normalized by DESeq2 and fold-change shrinkage was applied by setting the parameter "betaPrior=TRUE". ClusterProfiler[77] version 3.12.0 was used to perform functional enrichment analyses based on Kyoto Encyclopedia of Genes and Genomes (KEGG) pathways and Gene Ontology (GO) terms. The enricher function was used to perform hypergeometric tests based on lists of significant genes and the GSEA function was applied for gene set enrichment analysis considering the DESeq2 log2FoldChange of all analyzed genes.

## Puromycylation

Motoneurons were cultured for 5 DIV on laminin-111-coated glass coverslips. Cells were incubated with medium containing 10 μg/ml puromycin (Sigma-Aldrich, P8833) for 15 min at 37 °C in a humidified incubator. In negative control experiments, cells were pre-treated with 100 μg/ml cycloheximide for 30 min prior to the addition of puromycin to the medium. Cells were washed twice with prewarmed HBSS and fixed for 15 min in 4% paraformaldehyde (PFA) (Thermo Fisher Scientific, 28908). After fixation, cells were washed three times with PBS and permeabilized with 0.2% Triton X-100 for immunofluorescence staining using antibodies against puromycin and tau.

## Proximity ligation assay (PLA)

PLA was performed using the Duolink In Situ Orange Starter Kit Mouse/Rabbit (Sigma-Aldrich, DUO92102) according to the manufacturer's instructions. Briefly, motoneurons were cultured from *Hnrnpr*[−/−] and +/+ mice for 5 DIV on laminin-111-coated glass coverslips and washed twice with PBS. Motoneurons were fixed in paraformaldehyde lysine phosphate (PLP) buffer (4% PFA, 5.4% glucose and 0.01 M sodium metaperiodate) for 10 min. After permeabilization and washing, motoneurons were blocked in a blocking buffer for 1 h at 37 °C. Cells were incubated with primary antibodies diluted in a blocking buffer overnight at 4 °C. PLA probes were applied at 1:5 dilution for 1 h at 37 °C followed by ligation for 30 min at 37 °C and amplification for 100 min at 37 °C. Cells were fixed again for 10 min at room temperature in PLP buffer, washed with PBS, and processed further for immunofluorescence and 4′,6-diamidino-2-phenylindole (DAPI) staining.

For Puro-PLA, motoneurons were incubated with medium supplemented with 10 μg/ml puromycin for 8 min at 37 °C in a humidified incubator. Motoneurons were washed twice with prewarmed HBSS and fixed for 10 min in PLP buffer. After fixation, cells were washed three times with PBS, permeabilized and processed for PLA using antibodies against puromycin (Sigma-Aldrich, MABE343, 1:200 dilution) and Macf1 (Abcam, ab117418, 1:100 dilution).

## Immunofluorescence staining

Motoneurons were cultured on laminin-111-coated glass coverslips for 5-6 DIV. Cells were washed three times with PBS and fixed with 4% PFA at room temperature for 15 min followed by permeabilization with 0.2% Triton X-100 at room temperature for 20 min. After three washes in PBS, cells were blocked in a blocking buffer containing 4% donkey serum at room temperature for 1 h. Primary antibodies diluted in blocking solution were applied onto coverslips and incubated at 4 °C overnight followed by incubation with secondary antibodies at room temperature for 1 h and counterstaining with DAPI. Coverslips were washed three times with PBS, mounted using FluorSave Reagent (Merck, 345789) and subsequently imaged on an Olympus confocal microscope FV 1000 confocal system at 60× magnification. Intensity and region-of-interest analyses were performed using ImageJ as part of the Fiji package[78]. Antibodies are listed in Supplementary dataset 11.

## Immunohistochemical staining of neuromuscular junctions

The tibialis anterior (TA) muscles of *Hnrnpr*[−/−] and +/+ were fixed in 4% PFA for 2 h at room temperature. After washing with PBS, the muscle tissue was incubated with 0.1 M glycine (Roth, 3790) for 30 min and then permeabilized with 0.5% Triton X-100 for 1 h. For blocking, muscle tissue was incubated in a blocking solution containing 4% donkey serum and 0.3% Triton X-100 in PBS for 2 h at room temperature. Antibodies against Synaptophysin 1 (Synaptic Systems, 101004, 1:500 dilution), Neurofilament heavy chain (EMD Millipore, AB5539, 1:1000 dilution) and choline acetyltransferase (EMD Millipore, AB144P, 1:200 dilution) diluted in blocking solution were applied for 48 h at 4 °C. Thereafter, the muscles were washed three times with PBS containing 0.01% Tween-20 (PBS-T) for 10 min and incubated with α-bungarotoxin conjugated to Alexa Fluor 488 (1:500) to label the postsynaptic part of neuromuscular endplates and secondary antibodies for 1 h at room temperature followed by three washes with PBS-T. Then, muscles were DAPI-stained, mounted onto object slides, and embedded with FluorSave Reagent. Antibodies are listed in Supplementary dataset 11.

## Immunohistochemical staining of spinal cord sections

Mice were deeply anesthetized and transcardially perfused with 0.1 M phosphate buffer (PB) (pH7.4) and 4% PFA in PB. Spinal cords were dissected and post-fixated for 2 h in PFA at room temperature. Samples were washed, embedded in 6% agarose cubes, and cut into 40 μm

sections using a vibratome (Leica, VT1000S). For antibody labeling, sections were washed with PBS, then immersed in ammonium acetate buffer (50 mM ammonium acetate, 2 mM CuSo4, PH:5) for 30 min and washed again in PBS. Sections were quenched with 0.1 M glycine for 15 min and blocked with 4% donkey serum and 0.3% Triton X-100 in PBS for 2 h at room temperature, and subsequently incubated with the primary antibodies at 4 °C for 48 h in blocking solution. After washing three times with PBS containing 0.3% Triton X-100, sections were incubated with secondary antibodies for 2 h at room temperature and washed again three times with PBS containing 0.3% Triton X-100. Then, sections were DAPI-stained, mounted onto object slides, and embedded with FluorSave Reagent. Antibodies are listed in Supplementary dataset 11.

## Image acquisition and data analysis

Images were acquired on an Olympus FV 1000 confocal system equipped with the following objectives: 10× (NA: 0.25), 20× (NA: 0.75), 40× (oil differential interference contrast, NA: 1.30), or 60× (oil differential interference contrast, NA: 1.35). Fluorescence excitation was achieved by using 405, 473, 559, and 633 nm lasers. Images were obtained with the corresponding Olympus FV10-ASW imaging software for visualization. The resulting images (Olympus.oib format) were processed by maximum intensity projection and were adjusted for brightness and contrast using Image J software. For intensity measurements, mean gray values of images were measured from unprocessed raw data after background subtraction using ImageJ software.

For intensity measurements of eS6-eL24 and RL2-eIF4G PLA signals in the somata, mean gray values of images were measured from unprocessed raw data after background subtraction using ImageJ software. For Pabp-eIF4G PLA and Macf1 Puro-PLA signal in the somata and in the 50 μm-long proximal axonal regions, the number of punctae was quantified semi-automatically using the ImageJ threshold and particle analysis plugin after background subtraction.

For intensity measurements of puromycin, raw images were projected using ImageJ and mean gray values were measured after background subtraction. Puromycin intensity was measured in somata and 20 μm-long proximal and distal regions of axons.

For axon length measurements, cultured *Hnrnpr*[−/−] and +/+ motoneurons were immunostained at DIV 2, 4, and 6 with anti-tau antibody. The images were acquired with a Keyence BZ-8000K fluorescence microscope equipped with a standard color camera using a 20× 0.7-NA objective. The length of the longest axon branch was measured using ImageJ software. Motoneurons were only scored when designated axons were at least three times longer than the corresponding dendrites ensuring an unambiguous distinction between axons and dendrites[79]. Spinal cord sections were imaged on a fluorescence microscope (AxioImager 2, Zeiss) and z-stack images were adjusted for brightness and contrast using ImageJ software. anti-Iba1 and ChAT immunosignals were quantified semiautomatically using the ImageJ threshold and particle analysis plugin after background subtraction.

## Protein extraction and western blotting

Total protein was extracted from brain tissue with RIPA buffer (50 mM Tris-HCl pH7.4, 150 mM NaCl, 1% NP-40, 0.05% sodium deoxycholate, 0.1% SDS). Protein concentration was measured using a BCA protein assay kit (Thermo Fisher Scientific, 23227). Equal amounts of proteins were size-separated by SDS-PAGE gel electrophoresis followed by transfer onto nitrocellulose membrane and immunoblotting with the indicated antibodies (Supplementary dataset 11).

## Sucrose gradient fractionation

$2 \times 10^7$ primary motoneurons were plated in a laminin-111-coated microfluidic chamber and cultured for 7 DIV. The somatodendritic

and the axonal compartment were pretreated with 100 μg/ml cycloheximide for 10 min at 37 °C. Cells were then lysed in polysome lysis buffer (20 mM Tris-HCl pH 7.4, 100 mM KCl, 5 mM MgCl$_2$, 1 mM Leupeptin, 1 mM Pepstatin, 1 mM Aprotinin, 0.1 mM PMSF, 0.1 mM AEBSF, 0.5% NP-40, 100 μg/ml cycloheximide, protease inhibitor, 40 U/ml RNAsin (Promega)). After 10 min incubation on ice, lysates from both somatodendritic and axonal compartments were centrifuged at 10,000 × *g* for 10 min at 4 °C. 20 μl of cleared lysate were kept as input sample and the remaining lysate was loaded onto 5–45% sucrose gradients in gradient buffer (20 mM Tris-HCl pH 7.5, 100 mM KCl, 5 mM MgCl$_2$) and centrifuged at 34,500 rpm for 2 h at 4 °C in a SW60 swing-out rotor (Beckmann Coulter). Gradients were harvested with a Biocomb PGFip Piston Gradient Fractionator that continually monitored the optical density at 254 nm. For western blot analysis, proteins were precipitated using methanol/chloroform precipitation[80]. Antibodies used and their dilutions are listed in Supplementary dataset 11.

### Thiamet-G and OSMI-1 treatment

For immunostaining and PLA, cultured motoneurons from *Hnrnpr*$^{-/-}$ and +/+ mice were treated on DIV 5 with TMG (Abcam, ab146193) at a concentration of 100 μM or with OSMI-1 (Abcam, ab235455) at a concentration of 50 μM for 24 h followed by fixation. For axon length measurements, cultured motoneurons were treated on DIV 5 with 100 μM TMG for 24 h (single dose) and then cultured until DIV 6. For two doses, motoneurons were treated on DIV 4 and 5 with 100 μM TMG for 24 h each. Motoneurons were fixed on DIV 6 and assessed for axon length as described above.

### Grip strength measurements

Force measurements were performed on a metal triangle detecting forces within a range of 0–200 cN. Animals were first trained to hold on the grid, and, after they had grasped the grid using both paws, a force was applied to pull the mice away from the grid. The force required to detach the mice from the grid was recorded, and the mean value from five attempts was taken as one data point and considered as the grip strength.

### Rotarod

For testing, the RotaRod was accelerated from 10 to 40 rpm in 20 s in one direction and then again in the opposite direction and repeated until 60 s were reached. Animals were assayed on the RotaRod monthly, and performance over 16 months was reported in monthly time bins.

### Statistics and reproducibility

All statistical analyses were carried out using GraphPad Prism version 9 for Windows (GraphPad Software, San Diego, California USA), and statistical significance was determined at test level $P < 0.05$. Quantitative data were expressed as mean ± s.d. from at least three independent experiments unless otherwise indicated. No statistical method was used to predetermine the sample size. No data were excluded from the analyses. Two groups were compared using unpaired two-tailed Student's *t* test or two-tailed one-sample *t*-test. For multiple independent groups, one-way or two-way analysis of variance (ANOVA) with a post hoc multiple comparisons test was used. For box plots, boxes represent the interquartile range (IQR), which is the range between the first (Q1) and the third (Q3) quartile, lines inside the boxes represent the median of the data and whiskers extend from the box to the smallest and largest values within 1.5 times IQR from Q1 and Q3, respectively. Points outside the whiskers were considered outliers and were plotted individually. When the data did not adhere to a normal distribution, statistical analysis was performed using SuperPlots[81]. Details of replicate numbers, quantification, and statistics for each experiment are specified in the figure legends.

### Reporting summary

Further information on research design is available in the Nature Portfolio Reporting Summary linked to this article.

## Data availability

Source data are provided with this paper. The raw mass spectrometry proteomics data have been deposited to the ProteomeXchange Consortium via the PRIDE partner repository as three datasets. One dataset contains the total proteome data of axons and soma and has the identifier PXD038368. Another dataset contains the hnRNP R interactome data and has the identifier PXD043851. The third dataset contains the OGT and RL2 interactome data and has the identifier PXD052115. The RNA-seq data have been deposited at Gene Expression Omnibus (GEO) under accession number GSE242027. Source data are provided with this paper.

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

## Acknowledgements

The authors thank the Core Unit SysMed at the University of Wuerzburg for excellent technical support, RNA-seq data generation, and analysis. The Core Unit SysMed is supported by the IZKF at the University of Wuerzburg (project Z-6). We thank Flinders University for providing the p75NTR antibody. This work was supported by the Deutsche Forschungsgemeinschaft [BR4910/1-2 (SPP1738) and BR4910/2–2 (SPP1935) to M.B., SE697/4-2 (SPP1738) and SE697/5–2 (SPP1935) to M.S., Fi573/15-2 (SPP1935) and Fi573/20-1 to U.F.] and a Grant from the Schilling Stiftung im Stifterverband für die Deutsche Wissenschaft to M.S.

## Author contributions

M.B., M.S., and A.Z. designed the study. A.Z., S.S., and J.B. performed the experiments. A.Z. and S.S. conducted statistical analysis and interpretation of the data. J.B. performed proteomics under supervision of M.M. C.S. performed sucrose density gradient ultracentrifugation under supervision of U.F. A.V. contributed to muscle preparation. P.A. performed RNA-seq. A.Z. and S.S. wrote the manuscript. M.B. and M.S. reviewed and revised the manuscript. M.S. and M.B. supervised this study and provided financial support. All authors approved the final version of the manuscript.

## Funding

## Competing interests

The authors declare no competing interests.
