## [Peer Review File · Nature Communications]

hnRNP R promotes O-GlcNAcylation of eIF4G and facilitates axonal protein synthesisREVIEWER COMMENTS

Reviewer #1 (Remarks to the Author):

In his study, Zare and colleagues present a hnRNP R knockout mouse that exhibits reduced axonal lengths without a significant impact on lifespan. They culture the motor neurons in microfluidic chambers to separate neurites from somata and employ various analytical methods, including mass spectrometry, RNA-seq, IP and fluorescence microscopy. Based on these approaches, the authors conclude that hnRNP R facilitates initiation by directly interacting with the Ogr, an acetylglucosamine transferase that O-GlcNAc-ylates eIF4G. While the data are convincing that hnRNP R impairs axonal growth, the bids presented as evidence for direct impact on initiation appear less compelling. hnRNP R is typical RNA-binding protein important component of several granules including neuronal ones. One plausible alternative explanation could be that the hnRNP R knockout alters the translocation/transport of some mRNAs into axonal ends, indirectly causing a translation deficiency. The authors should conduct a more rigorous analysis of their data in addition to employing more precise approaches to establish a clear association between hnRNP R and the translation machinery. The following points highlight several notable issues with the results:

1. hnRNP R knockout alters ribosomal association of Macf1 in neurites but not in somatodendrites. It is essential to investigate whether this effect is due to altered amounts of Macf1 transcript or altered amounts of ribosomes. Polysome profiles should be performed to measure the amount of Macf1 and ribosomes in both $+/+$ and $-/-$ mice in each fraction (neurites vs. somata). Along with this, the amounts of the ribosomes in each part of the motor neurons should be established. Maybe the authors already have this evidence from their mass spectrometry and the RNA-seq data sets.

Rigorous quantification of RNA amounts following hnRNP R depletion is crucial, potentially achieved by spiking libraries with standards to exclude the impact of hnRNP R depletion on mRNA transport and supply in axons.

2. Fig. 3: To assess translational influence of hnRNP R, the authors compare proteomics with RNA-seq, missing, however, the most crucial step in the production of the proteins – translation. Ideally, the authors should utilize ribosome profiling, or as a less expensive substitute, polysome profiling combined with RNA-seq to demonstrate which mRNAs are actively translated, and whether the hnRNP R depletion alters this.

3. Fig 4a: The hnRNP R signal disappears following RNaseI treatment., suggesting hnRNP R binding to mRNA rather than direct association with the ribosome. The authors should use the published CLIP-data (they use) to show hnRNP R binding peaks (aggregated plot across all mRNAs bound to hnRNP R). Caution is needed in interpreting Fig.4e. The co-migration of eIF2a and hnRNP R in the sucrose gradients, does not necessarily imply a direct association! Both proteins might be associated with different mRNA clients

The gel on Fig. 4g caught reviewers attention as a likely candidate for potential flow artifacts. The inputs of eIF2a and eIF4G look identical, which even by loading identical sample would unlikely produce an identical shape on the blots. Also, hnRNP R-blot sees flipped. In accordance with all other data, shouldn't hnRNP R be the highest in the input fraction?

Treatment with benzonase does not seem appropriate. The choice should be RNase I, as in Fig. 4a.

4. From Fig. 5 the conclusion about the direct participation of hnRNP R in ribosomal assembly lack direct evidence. Microscopy, albeit illustrative, is not a quantitative approach. The authors have mass spec data and should use them for quantitative assessment of the ribosomes. Ribosomal proteins, the easiest detectable in mass spec, are in equimolar amounts in the ribosomes.

5. In Fig 6b, the $-/-$ mouse is missing.

6. In the introduction, the literature review on local translation in neurons, particularly the pioneering work from the group of Erin Schuman, is lacking, giving the impression that local translation has not been studied at all. Incorporate relevant literature to provide context.

Minor

1. Please use the new nomenclature for labeling ribosomal protein. It has been in use for almost 10 y already (PMID: 24524803)
2. Fig. 4d: The polysome profiles are poor and give an impression as no active ribosomes are present. Remove the first fraction of unbound RNA in the profiles to enhance clarity and better showcase active ribosomes. (i.e. the lower intensity fractions 3-12). For polysome profiles, consider higher amounts when possible and add cycloheximide in the polysome buffer for stabilization.

Reviewer #2 (Remarks to the Author):

The manuscript by Zare et al. provides a comprehensive exploration of how hnRNP R serves as a central player in the O-glcNAcylation of eIF4G within axons, contributing to enhanced local translation. The findings presented in this study are intriguing and contribute interesting insights into the localization and translation processes within axons, appealing to the broad readership in Nature Communications. This reviewer recommends addressing the following points before the publication.

Major points:

1. The authors would not elucidate or discuss why hnRNP R exhibits a more specific function in axonal translation than the somatodendrite. While whole experiments in this paper show the significance of hnRNP R in axonal protein translation, the critical question remains unanswered: why the strong impact of hnRNP R was observed in the axon specific, despite its presence in both the axon and the cell body? A more comprehensive explanation from the authors is needed.
2. Although this reviewer is not sure whether this is a reasonable request, the application of ribosome profiling would be helpful for the investigation of the specificity of hnRNP R's role in somatic cells and axons, respectively. This will clarify the mRNA targets by hnRNP R.
3. Related to point 1, the authors did not explain why hnRNP R regulates Ogt transferase in an axon-specific manner. Considering that Ogt expressed ubiquitous in neurons, hnRNP R could also regulate in somato, in theory. A detailed rationale and/or experimental evidence should be provided.
4. Throughout the paper, there needed to be more explanation for the quantification methods used in immunostaining, PLA, and puroPLA. The authors should consider providing the details, including how many fields or cells were counted and how many replicates were conducted.
5. In lines 271-272, the authors jumped to the link to eIF4G. Although eIF4G is known to have O-GlcNAcylation, there is no rationale to focus on the modification of eIF4G (since other initiation factors may have O-GlcNAcylation). The authors should provide straightforward logic/data to focus on eIF4G in the corresponding sentences.
6. In Fig. 6e, the author employed PLA experiments to confirm the GlcNAc modification of eIF4G. However, in theory, this experiment could not distinguish 1) GlcNAc modification on eIF4G and 2) GlcNAc modification on other proteins closely located with eIF4G. The authors should use alternative (more direct) methods to assess O-GlcNAc modification on eIF4G.
7. The authors should test whether the eIF4G-PABP interaction found in the axon was dependent on eIF4G GlcNAc modification.
8. Regarding Fig. 5a, the data were based solely on PLA evidence. It would be beneficial to ensure these findings by conventional western blot analyses.

9. Taking together the data in this study, the model in Fig. 5g, may not be accurate, although this reviewer understood that this cartoon is intended to summarize the results up to Fig. 5. Given the hnRNP R enhances Ogt-mediated eIF4G O-GlcNAc for PABP interaction and translation activation, placing the hnRNP R as a part of the complex (not highlighting the above points) may be confusing to readers. The authors should consider to re-drawn the model to fit the current results.

10. Once the experimental setup in Fig. 6f could monitor O-GlycNAcylation of eIF4G, the data showed the TMG treatment also increased O-GlycNAcylation of eIF4G in soma as well. This reviewer wonders why this did not lead to translation activation in soma (Fig. 6g).

Minor points:

1. At lines 219-219, given the data provided in Fig. 4a-g, it was quite hard to specify that hnRNP R's regulatory point lies at translation initiation. The authors should tone down this claim.

2. Regarding Fig. 5a, it would be straightforward to speculate that the hnRNP R-mediated eIF4G-PABP interaction enhancement is based on the O-GlycNAcylation of eIF4G. The authors should mention this interpretation in the text.

3. The authors may consider moving Figures 1a, b, d, e, and f (concerning KO validation) to the Extended Figure section.

4. The custom gene ontology category of "cytoskeleton" used in Fig. 2h-i, should be explained in detail in the method section.

5. It should be helpful to denote the "Ogt" in the left panel of Fig. 6a (Somatodendritic panel). Related to this, it would be informative to perform IP-western experiments between Ogt and hnRNP R in both somatodendritic and axon fractions.

6. Typo in Fig. 1g:

"DAPI/Tubulin/hnRPN R" should be "DAPI/Tubulin/hnRNP R"

7. For Fig. 2h, the authors should consider a cumulative fraction plot or box plot for better visualization and statistical analysis.

8. Regarding Fig. 6a, the authors should directly compare the interaction strength of each protein found in somatodendritic and axon fractions. This reviewer wonders whether a general correlation could be found between the two data sets.

Reviewer #3 (Remarks to the Author):

In the manuscript entitled "hnRNP R promotes O-GlcNAcylation of eIF4G to facilitate axonal protein synthesis", Zare and colleagues investigate the role of the RNA-binding protein hnRNP R in local axonal protein synthesis and its effect on motoneuron function. To this end, they generated a new Hnrnpr

knockout mouse, providing them the opportunity to investigate functions in vivo. Using this approach, they validated their previous findings that hnRNP R depletion affects axonal outgrowth. Using a range of imaging and 'omics' methods, the authors uncover that hnRNP R deficiency leads to perturbed local protein synthesis and an altered axonal proteome. Furthermore, the authors uncover O-GlcNAcylation as a mechanism by which translation initiation is likely affected by hnRNP R via its regulation of the O-GlcNAc transferase. Though experiments in living mice are limited to a few initial readouts, they confirm that the axon outgrowth defects observed in the current and previous studies indeed lead to motor defects. In summary, the presented data nicely validates and expands upon the authors' previous publications. The experiments are generally well-designed, the manuscripts well-written, and the data presented in clear graphs. Some relatively minor inquiries remain.

- For Fig. 1h: could the authors please specify how survival was assessed in the methods?
- The authors should specify not only the number of biological replicates but also the number of axons quantified per replicate (e.g. Fig. 1j) in the figure legend for all relevant graphs.
- Though it is intuitive, in Fig. 2i could the authors state what is encoded by the size of the circle in the figure legend.
- The title in line 166 states proteome alterations are not due to mRNA transport defects. Though this is a likely interpretation, this is not directly shown by the data or further specified in the following paragraphs. Therefore, I would ask the authors to adapt the title and elaborate on this topic rather in the discussion. Is there an overlap between targets regulated in axons and the somatodendritic compartment? Out of curiosity, would the observation that downregulated targets in axons contain more iCLIP sites (Fig. 3d) also be true for the somatodendritic compartment?
- The authors should clearly state in the text what each figure shows. For instance, explain that Extended data 3a is the control for the mentioned data.
- In Fig. 6a, is Ogt also enriched in the somatodendritic compartment? Could the authors indicate Ogt in the somatodendritic compartment?
- It is indeed interesting that TMG treatment can restore O-GlcNAc levels (Extended Data Fig. 4). However, would it not be more informative to attempt to rescue O-GlcNAc levels by reintroducing hnRNP R. This would validate the direct involvement of hnRNP R which is the final interpretation of this paragraph.
- In line 341 in the discussion, the author state that translation was assessed in the NMJ. Could the author point out this data, as we might have missed it in the manuscript.
- Perhaps the authors could mention/discuss the relevance of Ptbp2 for hnRNP R and the current data as discovered in their previous publication.
- The figures are partially somewhat crowded, and specifically microscopy images are quite small, which may be even more drastic in a final layout. Perhaps the authors could adjust image size, text size, etc. so the data can be easily seen in a final layout.

Reviewer #1 (Remarks to the Author):

In his study, Zare and colleagues present a hnRNP R knockout mouse that exhibits reduced axonal lengths without a significant impact on lifespan. They culture the motor neurons in microfluidic chambers to separate neurites from somata and employ various analytical methods, including mass spectrometry, RNA-seq, IP and fluorescence microscopy. Based on these approaches, the authors conclude that hnRNP R facilitates initiation by directly interacting with the Ogr, an acetylglucosamine transferase that O-GlcNAc-ylates eIF4G. While the data are convincing that hnRNP R impairs axonal growth, the bids presented as evidence for direct impact on initiation appear less compelling. hnRNP R is typical RNA-binding protein important component of several granules including neuronal ones.

One plausible alternative explanation could be that the hnRNP R knockout alters the translocation/transport of some mRNAs into axonal ends, indirectly causing a translation deficiency. The authors should conduct a more rigorous analysis of their data in addition to employing more precise approaches to establish a clear association between hnRNP R and the translation machinery. The following points highlight several notable issues with the results:

1. hnRNP R knockout alters ribosomal association of *Macf1* in neurites but not in somatodendrites. It is essential to investigate whether this effect is due to altered amounts of *Macf1* transcript or altered amounts of ribosomes.

Author's response: We agree with the reviewer that alterations in mRNA binding to ribosomes could also be a consequence of altered mRNA levels. To rule out this possibility, we measured the level of *Macf1* mRNA in somatodendritic and axonal compartments of *HnrnpR*^{-/-} and +/+ motoneurons cultured in microfluidic chamber by qPCR. The results show that the level of *Macf1* mRNA was not changed in *HnrnpR*^{-/-} motoneurons in either compartment (Fig. 3d in the revised manuscript). In agreement with these results, our RNA-seq data from compartmentalized *HnrnpR*^{-/-} motoneurons similarly revealed that the axonal and somatodendritic level of *Macf1* mRNA remained unchanged (Supplementary Table 5 and 6 in the revised manuscript). Therefore, the reduced association of *Macf1* with ribosomes in axons of *HnrnpR*^{-/-} motoneurons is not due to lower *Macf1* mRNA levels. In order to clarify this point, we updated the manuscript and included the *Macf1* data as a new main figure (Fig. 3 in the revised manuscript).

According to our proteomics data, 61 ribosomal proteins were detected in the axonal compartment, of which two ribosomal proteins (eS2 and eL13) were upregulated and 5 ribosomal proteins (eS18, eS3, eL14, eL15, and eS15a) were downregulated upon loss of hnRNP R (Supplementary Table 3). This further indicates that ribosomes in general are not depleted in axons of *Hnrnp1r*^{-/-} motoneurons. We updated the text of the manuscript to include this point.

Polysome profiles should be performed to measure the amount of Macf1 and ribosomes in both +/+ and -/- mice in each fraction (neurites vs. somata).

Author's response: We agree with the reviewer that performing polysome profiling in *Hnrnp1r*^{-/-} and +/+ motoneurons would be a valuable experiment to obtain further insights how hnRNP R controls *Macf1* translation. However, since *Hnrnp1r*^{-/-} mice are derived from crosses of *Hnrnp1r*^{+/-} mice, representing 25% of the offspring, this approach is technically not feasible due to the limited amounts of axonal material that can be obtained from motoneurons grown in compartmentalized chambers. From one mouse embryo, about 20,000 spinal motoneurons can be isolated. Polysome profiling is usually done with a minimum of 1,000,000 cells, and since axons contain less than 5% of the cytoplasm of motoneurons, at least 1,000 embryos would be necessary for one single experiment. For this reason, we performed Y10b immunoprecipitation as a way to provide evidence for reduced *Macf1* mRNA association with ribosomes in axons of *Hnrnp1r*^{-/-} motoneurons (Fig. 3c in the revised manuscript). Additionally, our Puro-PLA analysis showed reduced nascent synthesis of Macf1 protein in axons of *Hnrnp1r*^{-/-} motoneurons (Fig. 3a,b in the revised manuscript).

Along with this, the amounts of the ribosomes in each part of the motor neurons should be established. Maybe the authors already have this evidence from their mass spectrometry and the RNA-seq data sets.

Author's response: We followed up on the reviewer's suggestion and inspected our proteomics data from somatodendritic and axonal compartments of *Hnrnp1r*^{-/-} and +/+ motoneurons. According to our data, 61 ribosomal proteins were detected in axonal compartments, of which two ribosomal proteins (eS2 and eL13) were upregulated and 5 ribosomal proteins (eS18, eS3, eL14, eL15, and eS15a) were downregulated upon loss of hnRNP R (Supplementary Table 3). We also detected

95 ribosomal proteins in the somatodendritic compartment and, of these, 16 and 11 ribosomal proteins were upregulated and downregulated, respectively, in hnRNP R-depleted motoneurons (Supplementary Table 2). We found that among deregulated ribosomal proteins, only eS2 was elevated in both compartments. These data thus indicate that there is no general depletion of ribosomes in axons of *Hnrnp1r*^{-/-} motoneurons. We have now included this information in the revised manuscript. These data also provide evidence that the altered axonal translation is not due to reduced levels of ribosomal components in the axons of *Hnrnp1r*^{-/-} motoneurons.

Rigorous quantification of RNA amounts following hnRNP R depletion is crucial, potentially achieved by spiking libraries with standards to exclude the impact of hnRNP R depletion on mRNA transport and supply in axons.

Author's response: We agree with the reviewer that RNA quantification is an important quality control step prior to RNA-seq. Therefore, we evaluated the amounts and integrity of the RNA from the somatodendritic compartments on a 2100 Bioanalyzer:

Samples 5-8 were from *Hnrnp1r*^{+/+} motoneurons and samples 9-12 were from *Hnrnp1r*^{-/-} motoneurons (samples 1 and 2 were test samples from *Hnrnp1r*^{+/+} motoneurons). The concentrations of all samples were similar and their RIN values were between 8.2 and 9.6. The RNA amounts from axonal compartments were too low for Bioanalyzer analysis. Therefore, we used ¼ volume of the SMART-Seq® v4 Ultra® Low Input RNA Kit (Takara) and 10 PCR cycles for amplification of somatodendritic samples, and the full reaction volume and 15 PCR cycles for

amplification of axonal samples. Following amplification, dsDNA was quantified by Qubit™ dsDNA HS Assay Kit (3.0 Fluometer; Thermo Fisher Scientific) and quality was checked using the 2100 Bioanalyzer. Following RNA-seq library preparation by tagmentation, their amounts were analysed by Qubit and Bioanalyzer, and then pooled. The pooled RNA-seq libraries were spiked with 1% PhiX control library and then sequenced.

2. Fig. 3: To assess translational influence of hnRNP R, the authors compare proteomics with RNA-seq, missing, however, the most crucial step in the production of the proteins – translation. Ideally, the authors should utilize ribosome profiling, or as a less expensive substitute, polysome profiling combined with RNA-seq to demonstrate which mRNAs are actively translated, and whether the hnRNP R depletion alters this.

Author's response: In principle, we agree with the reviewer that performing polysome profiling on compartmentalized *Hnrnp^r*^{-/-} and +/+ motoneurons followed by RNA-seq could be useful for investigating the defects in translational regulation upon loss of hnRNP R. However, this approach is technically not feasible due to the limited amounts of axonal material that can be obtained from compartmentalized motoneurons, as stated above. One mouse embryo only contains maximally 20,000 motoneurons. Nevertheless, according to our proteomics results and, in particular, the data from the axonal puromycinylation assay (Fig. 2a,b in the revised manuscript), many proteins are deregulated in axons of *Hnrnp^r*^{-/-} motoneurons while the encoding mRNAs, as measured by RNA-seq, were not altered. This indicates that axonal translation, but less so mRNA transport or degradation, is regulated by hnRNP R. In our manuscript, we showed this in detail for *Macf1*, which is one of the most strongly downregulated proteins in axons of *Hnrnp^r*^{-/-} motoneurons. We performed ribosome immunoprecipitation using the Y10b antibody and found the reduced association of *Macf1* mRNA with ribosomes in axons of *Hnrnp^r*^{-/-} motoneurons (Fig. 3c in the revised manuscript). Additionally, our Puro-PLA analysis showed reduced nascent synthesis of *Macf1* protein in axons of *Hnrnp^r*^{-/-} motoneurons (Fig. 3a,b in the revised manuscript) while axonal *Macf1* mRNA levels were unchanged (Fig. 3d in the revised manuscript).

3. Fig 4a: The hnRNP R signal disappears following RNaseI treatment., suggesting hnRNP R binding to mRNA rather than direct association with the ribosome. The authors should use the published CLIP-data (they use) to show hnRNP R binding peaks (aggregated plot across all mRNAs bound to hnRNP R).

Author’s response: In our previous study (Briese et al. PNAS doi: 10.1073/pnas.1721670115), we plotted the enrichment of hnRNP R iCLIP hits in the different gene segments because the exact positioning of hnRNP R binding varies across mRNAs. This revealed enriched hnRNP R binding in 3’ UTRs in line with the function of hnRNP R for translation initiation we show in our current study:

Enrichment of hnRNP R iCLIP hits (from Briese et al. PNAS doi: 10.1073/pnas.1721670115).

As an example, we find hnRNP R iCLIP hits in the 3’ UTR of the *Macf1* mRNA, in line with the function of hnRNP R in regulating the local translation of *Macf1* in axons:

Caution is needed in interpreting Fig.4e. The co-migration of eIF2a and hnRNP R in the sucrose gradients, does not necessarily imply a direct association! Both proteins might be associated with different mRNA clients.

Author’s response: We agree with the reviewer that co-migration through a gradient does not imply association. Therefore, we have adjusted the text in the revised manuscript as follows:

“We observed that hnRNP R was detectable at higher level in the 40S, 60S, and monosome fractions compared to the polysome fractions in both somatodendritic and axonal compartments, using eIF2 α as a control for the distribution of ribosomal subunits and monosomes across the fractions (Fig. 5e).”

Nevertheless, we also show by immunoprecipitation that eIF2 α associates with hnRNP R (Fig. 5g in the revised manuscript), thus providing independent and more specific evidence for an association of hnRNP R and eIF2 α .

The gel on Fig. 4g caught reviewers attention as a likely candidate for potential flow artifacts. The inputs of eIF2 α and eIF4G look identical, which even by loading identical sample would unlikely produce an identical shape on the blots.

Author's response: We have provided an uncropped Western blot below showing that eIF4G and eIF2 α are separate.

Also, hnRNP R-blot sees flipped. In accordance with all other data, shouldn't hnRNP R be the highest in the input fraction?

Author's response: In Fig. 5g in the revised manuscript, we immunoprecipitated hnRNP R protein, therefore the signal is enriched and the signal intensity is higher in the hnRNP R IP lane than in the input lane. Please also see the uncropped Western blot below (shorter exposure of the blots shown above). In Fig. 5a in the revised manuscript, ribosomal RNAs were immunoprecipitated using Y10b antibody, and hnRNP R was co-precipitated.

Treatment with benzonase does not seem appropriate. The choice should be RNase I, as in Fig. 4a.

Author's response: In Fig. 5a in the revised manuscript (previous Fig. 4a), we investigated the association of hnRNP R with intact ribosomes by immunoprecipitating ribosomal RNAs with the Y10b antibody. Therefore, lysates were treated with RNase I to digest ribosome-associated mRNAs but not ribosomes themselves. However, in Fig. 5g in the revised manuscript (previous Fig. 4g), we investigated the RNA-dependence of the interaction between hnRNP R and other translation factors. For this purpose, we used Benzonase in order to degrade mRNA as well as ribosomal RNAs. In order to clarify this, we modified the text as follows:

“Following treatment of lysate with Benzonase to digest mRNA as well as rRNA, co-purification of eIF2α with hnRNP R was unperturbed showing that their association is RNA-independent (Fig. 5g).”

4. From Fig. 5 the conclusion about the direct participation of hnRNP R in ribosomal assembly lack direct evidence. Microscopy, albeit illustrative, is not a quantitative approach. The authors have mass spec data and should use them for quantitative assessment of the ribosomes. Ribosomal proteins, the easiest detectable in mass spec, are in equimolar amounts in the ribosomes.

Author's response: The PLA signal directly correlates with protein complex assembly, generating a signal when the two antibodies used are less than 15 to 40 nm apart. Many studies have shown that the PLA signal is a quantitative measure of protein associations (eg. Söderberg

et al. Nat Methods doi: 10.1038/nmeth947; Gustafsdottir et al. Anal Biochem doi: 10.1016/j.ab.2005.01.018; tom Dieck et al. Nat Methods doi: 10.1038/nmeth.3319). Therefore, we think that our Pabp-eIF4G and eS6-eL24 PLA experiments in Fig. 6 in the revised manuscript (previous Fig. 5) reflect ribosome assembly defects in axons of *Hnrnp1r*^{-/-} motoneurons. The reviewer suggests to use our mass spectrometry data to assess ribosome assembly. However, only protein levels are measured by mass spectrometry, not their interactions. Nevertheless, we inspected our proteomics datasets for ribosomal proteins (please also see our reply to question 1 above). We observed that 61 ribosomal proteins were detected in axonal compartments, of which two ribosomal proteins (eS2 and eL13) were upregulated and 5 ribosomal proteins (eS18, eS3, eL14, eL15, and eS15a) were downregulated upon loss of hnRNP R (Supplementary Table 3). We also detected 95 ribosomal proteins in the somatodendritic compartment and, of these, 16 and 11 ribosomal proteins were upregulated and downregulated, respectively, in hnRNP R-depleted motoneurons (Supplementary Table 2). We found that among deregulated ribosomal proteins, only eS2 was elevated in both compartments. These data, which we included in the revised manuscript, thus indicate that there is no general depletion of ribosomes in axons of *Hnrnp1r*^{-/-} motoneurons. However, from these data we cannot draw conclusions regarding ribosome assembly.

5. In Fig 6b, the -/- mouse is missing.

Author's response: We now included the hnRNP R-Ogt PLA data for *Hnrnp1r*^{-/-} motoneurons (previously Supplementary Data Fig. 4a) in Fig. 7g in the revised manuscript.

Fig. 7g

6. In the introduction, the literature review on local translation in neurons, particularly the pioneering work from the group of Erin Schuman, is lacking, giving the impression that local translation has not been studied at all. Incorporate relevant literature to provide context.

Author's response: We now included additional references and updated the Introduction of the revised manuscript as follows:

“Axon growth involves the localization and local translation of mRNAs in axons and growth cones to generate a local supply of cytoskeletal components and other proteins required for axon development (Dalla Cost et al. Nat Rev Neurosci doi: 10.1038/s41583-020-00407-7). Growth cone turning towards chemoattractive signals involves local synthesis of β -actin (Leung et al. Nat Neurosci doi: 10.1038/nn1775). In adult mouse brains, ribosomes are present in presynaptic compartments, contributing to the diversity of the local proteome (Hafner et al. Science doi: 10.1126/science.aau3644; Glock et al. PNAS 10.1073/pnas.2113929118; van Oostrum et al. Cell doi: 10.1016/j.cell.2023.09.028). Furthermore, there is evidence that ribosomes are locally remodelled in axons to fine-tune local protein production (Shigeoka et al. Cell Rep doi: 10.1016/j.celrep.2019.11.025; Fusco et al. Nat Commun doi: 10.1038/s41467-021-26365-x).”

Minor

1. Please use the new nomenclature for labeling ribosomal protein. It has been in use for almost 10 y already (PMID: 24524803)

Author's response: We have now labeled the ribosomal proteins based on the new nomenclature throughout the revised manuscript. For example:

“To validate our finding, we performed PLA in cultured *Hnrnp1*^{-/-} and +/+ motoneurons using antibodies against the ribosomal proteins eS6 and eL24 to evaluate ribosome assembly (Fig. 6c in the revised manuscript and Supplementary Fig. 3c). eS6 and eL24 are components of the 40S and 60S ribosomal subunits, respectively, and form an intersubunit bridge that is critical for 80S ribosome assembly”.

2. Fig. 4d: The polysome profiles are poor and give an impression as no active ribosomes are present. Remove the first fraction of unbound RNA in the profiles to enhance clarity and better showcase active ribosomes. (i.e. the lower intensity fractions 3-12).

Author's response: We thank the reviewer for the suggestion. We have updated the polysome profile in the revised manuscript as follows:

Fig. 5d

For polysome profiles, consider higher amounts when possible and add cycloheximide in the polysome buffer for stabilization.

Author's response: We have added cycloheximide in the polysome buffer as described in our Methods section "Sucrose gradient fractionation":

"Cells were then lysed in polysome lysis buffer (20 mM Tris-HCl pH 7.4, 100 mM KCl, 5 mM MgCl₂, 1mM Leupeptin, 1mM Pepstatin, 1mM Aprotinin, 0.1 mM PMSF, 0.1 mM AEBSF, 0.5 % NP-40, 100 µg/ml cycloheximide, protease inhibitor, 40 U/ml RNAsin (Promega))."

Reviewer #2 (Remarks to the Author):

The manuscript by Zare et al. provides a comprehensive exploration of how hnRNP R serves as a central player in the O-glcNAcylation of eIF4G within axons, contributing to enhanced local translation. The findings presented in this study are intriguing and contribute interesting insights into the localization and translation processes within axons, appealing to the broad readership in Nature Communications. This reviewer recommends addressing the following points before the publication.

Author's response: We appreciate the reviewer's recognition of our findings contributing to understand the mechanisms of axonal translation.

1. The authors would not elucidate or discuss why hnRNP R exhibits a more specific function in axonal translation than the somatodendrite. While whole experiments in this paper show the significance of hnRNP R in axonal protein translation, the critical question remains unanswered: why the strong impact of hnRNP R was observed in the axon specific, despite its presence in both the axon and the cell body? A more comprehensive explanation from the authors is needed.

Author's response: We also consider it as an important question why hnRNP R regulates translation specifically in axons in several ways, and we have put more emphasis in the revised version of our manuscript to address differences between functions of hnRNP R in somata and axons. First, according to our hnRNP R interactome data (Fig. 7a in the revised manuscript), Ogt is a major interactor of hnRNP R in axons but not in cell bodies. We now confirmed this specificity by performing proteomics analysis of Ogt interactomes obtained by immunoprecipitating Ogt from axonal and somatodendritic compartments (Fig. 7c in the revised manuscript). In agreement with the hnRNP R interactome data showing Ogt enrichment in axonal hnRNP R complexes, hnRNP R was enriched in the axonal Ogt interactome but not in the somatodendritic Ogt interactome. Second, our data show that hnRNP R deficiency affects O-GlcNAcylation of eIF4G in axons but not in cell bodies of motoneurons (Fig. 8e,f in the revised manuscript), and that hnRNP R re-expression rescues axonal eIF4G O-GlcNAcylation (Supplementary Fig. 5a,b). We now performed additional PLA experiments showing that the association of eIF4G with Pabp is reduced in axons but not cell bodies of *Hnrnp1r*^{-/-} motoneurons, and that this association can be rescued by TMG treatment (Fig. 8g,h in the revised manuscript). As O-GlcNAcylation of eIF4G

stimulates translation, this compartment-specific pathway could explain why translation is affected in axons but not cell bodies of *Hnrnp1r*^{-/-} motoneurons. Finally, we identified O-GlcNAcylated proteins in axons of *Hnrnp1r*^{-/-} and +/+ motoneurons by performing proteomics analysis of proteins immunoprecipitated with the RL2 antibody (Fig. 8a-c in the revised manuscript). Importantly, this analysis revealed that O-GlcNAcylation of eIF4G, but not eIF4A, is affected by hnRNP R deficiency, further supporting the notion that hnRNP R regulates axonal translation through facilitating eIF4G O-GlcNAcylation. To clarify this further, we adjusted the Discussion as follows:

“In addition to ribosomal proteins, we identified Ogt as a major interactor of hnRNP R in axons but not cell bodies of motoneurons. Ogt-mediated O-GlcNAcylation is prevalent in the brain, and many neuronal and synaptic proteins are modified by O-GlcNAc addition⁴⁴. We observed reduced O-GlcNAc levels in axons of *Hnrnp1r*^{-/-} motoneurons that could be restored by treatment with the Oga inhibitor thiamet-G. Importantly, O-GlcNAcylation of eIF4G, which promotes its binding to PABP and, thus, mRNA circularization, was reduced in hnRNP R-deficient axons. Thiamet-G treatment could rescue axonal O-GlcNAcylation of eIF4G, thereby stimulating eIF4G binding to Pabp and restoring axonal protein synthesis in hnRNP R-deficient motoneurons. This suggests that hnRNP R stimulates Ogt activity in axons, thereby enhancing translation through O-GlcNAcylation of eIF4G, which increases its binding to Pabp.”

2. Although this reviewer is not sure whether this is a reasonable request, the application of ribosome profiling would be helpful for the investigation of the specificity of hnRNP R's role in somatic cells and axons, respectively. This will clarify the mRNA targets by hnRNP R.

Author's response: We agree with the reviewer that performing polysome profiling in *Hnrnp1r*^{-/-} and +/+ motoneurons would be very useful to investigate the hnRNP R-dependent ribosome association of axonal mRNAs. However, since *Hnrnp1r*^{-/-} mice are derived from crosses of *Hnrnp1r*^{+/-} mice, representing 25% of the offspring, this approach is technically not feasible due to the low amount of axonal material that can be obtained from compartmentalized motoneuron cultures. From one mouse embryo, about 20,000 spinal motoneurons can be isolated. Polysome profiling is usually done with a minimum of 1,000,000 cells, and since axons contain less than 5% of the cytoplasm of motoneurons, at least 1,000 embryos would be necessary for one single experiment.

3. Related to point 1, the authors did not explain why hnRNP R regulates Ogt transferase in an axon-specific manner. Considering that Ogt expressed ubiquitous in neurons, hnRNP R could also regulate in somato, in theory. A detailed rationale and/or experimental evidence should be provided.

Author's response: The reviewer raises the critical question why hnRNP R regulates Ogt activity in axons but not somatodendritic regions of motoneurons. As explained above in more detail, we addressed this question in several ways. We observed that Ogt is not enriched in the hnRNP R immunoprecipitate from the somatodendritic compartment ($P = 0.153$) while it is highly enriched in the hnRNP R immunoprecipitate from axons (please see Fig. 7a in the revised manuscript). Thus, the association between hnRNP R and Ogt primarily takes place in axons. We now identified the Ogt interactomes in somatodendritic and axonal compartments and observed hnRNP R enrichment selectively in axonal Ogt complexes, in agreement with our hnRNP R interactome data (Fig. 7c in the revised manuscript). To strengthen this point, we now highlighted Ogt and hnRNP R in all interactome data sets (Fig. 7a and c in the revised manuscript). Next, we performed additional experiments to confirm the association of Ogt with hnRNP R in axons by immunoprecipitation of hnRNP R in somatodendritic and axonal compartments of motoneurons followed by Western blot. We observed a strong interaction of Ogt with hnRNP R in axons but not in the somatodendritic compartment (Fig. 7f in the revised manuscript). Together, these data show that hnRNP R forms specific subcomplexes with Ogt in axons of motoneurons that are capable of regulating O-GlcNAcylation of eIF4G locally.

4. Throughout the paper, there needed to be more explanation for the quantification methods used in immunostaining, PLA, and puroPLA. The authors should consider providing the details, including how many fields or cells were counted and how many replicates were conducted.

Author's response: For all immunostainings, PLA and Puro-PLA assays, we have now included the number of cells per replicate in the figure legends in the revised manuscript. The methods to quantify these signals is reported in detail in the Image acquisition and data analysis of the method section of the revised manuscript.

5. In lines 271-272, the authors jumped to the link to eIF4G. Although eIF4G is known to have O-GlycNAcylation, there is no rationale to focus on the modification of eIF4G (since other initiation factors may have O-GlycNAcylation). The authors should provide straightforward logic/data to focus on eIF4G in the corresponding sentences.

Author's response: We now revised the manuscript and added additional data to clarify the focus on eIF4G. According to a previous study (Li et al. PNAS doi: 10.1073/pnas.1813026116), Ogt is part of the eIF4F complex and O-GlcNAcylates eIF4A and eIF4G. O-GlcNAcylation of eIF4G on Ser61 stimulates translation by promoting its interaction with PABP. Vice versa, O-GlcNAcylation of eIF4A disrupts the assembly of translation initiation complexes. We show that depletion of hnRNP R impairs translation in axons of cultured motoneurons (Fig. 2a). Therefore, we focused on eIF4G rather than eIF4A O-GlcNAcylation. Nevertheless, as an unbiased approach, we performed mass spectrometry analysis on axonal O-GlcNAcylated proteins following immunoprecipitation with the RL2 antibody. We observed that the O-GlcNAcylation of eIF4G, but not of eIF4A, was reduced in axons of *Hnrnp1*^{-/-} motoneurons and could be rescued by TMG treatment (Fig. 8a-c in the revised manuscript). Furthermore, we performed RL2 immunoprecipitation followed by immunoblot analysis for eIF4G and eIF4A (Fig. 8d in the revised manuscript). We observed that eIF4G, but not eIF4A, was O-GlcNAcylated in axons.

Fig. 8

6. In Fig. 6e, the author employed PLA experiments to confirm the GlcNAc modification of eIF4G. However, in theory, this experiment could not distinguish 1) GlcNAc modification on eIF4G and 2) GlcNAc modification on other proteins closely located with eIF4G. The authors should use alternative (more direct) methods to assess O-GlcNAc modification on eIF4G.

Author's response: We addressed the reviewer's comment in several ways. First, we investigated the literature to identify which other components of the eIF4F complex, which eIF4G is an integral part of, are potentially modified by Ogt in axons of motoneurons. According to two studies (Li et al. PNAS doi: 10.1073/pnas.1813026116; Wells et al. Mol Cell Proteom doi: 10.1074/mcp.M200048-MCP200) eIF4A, in addition to eIF4G, is O-GlcNAcylated. Therefore, to identify in an unbiased manner which axonal proteins depend on hnRNP R for O-GlcNAcylation, we immunoprecipitated O-GlcNAc-modified proteins using RL2 from somatodendritic and axonal lysate of compartmentalized motoneurons treated with DMSO (Ctrl) and TMG. We observed the O-GlcNAcylation of eIF4G, but not of eIF4A, was reduced in axons of *Hnnpnr^{-/-}* motoneurons and could be rescued by TMG treatment (Fig. 8a-c in the revised manuscript). Additionally, we performed immunoblot analysis on RL2 immunoprecipitates (Fig. 8d in the revised manuscript). We observed an immunosignal for eIF4G but not for eIF4A in axons, further supporting the notion that eIF4G is the main Ogt target in the eIF4F complex in axons.

Fig. 8

7. The authors should test whether the eIF4G-PABP interaction found in the axon was dependent on eIF4G GlcNAc modification.

Author's response: We addressed the reviewer's question in two ways. First, we performed eIF4G-Pabp PLA on *Hnnpnr^{-/-}* and +/+ motoneurons treated with DMSO (Ctrl) and the Oga inhibitor TMG (Fig. 8g,h in the revised manuscript). We observed that the reduced association between eIF4G and Pabp in axons of *Hnnpnr^{-/-}* motoneurons could be rescued by TMG treatment. Second, we treated motoneurons with the Ogt inhibitor OSMI-1 and observed a reduced eIF4G-Pabp PLA signal in axons but not cell bodies of motoneurons (Supplementary Fig. 5d,e in the revised

manuscript). Together with our data showing that eIF4G O-GlcNAcylation is reduced in axons of *Hnrnp1r*^{-/-} motoneurons (Fig. 8e,f in the revised manuscript) and that eIF4G but not eIF4A is O-GlcNAcylated in axons (Fig. 8a-d in the revised manuscript), these data further indicate that the eIF4G-Pabp interaction in axons is dependent on eIF4G O-GlcNAcylation.

Fig. 8

Supplementary Fig. 5

8. Regarding Fig. 5a, the data were based solely on PLA evidence. It would be beneficial to ensure these findings by conventional western blot analyses.

Author's response: The data in Fig. 5a (now Fig. 6a in the revised manuscript) contain PLA data showing reduced eIF4G-Pabp interaction in axons of *Hnrnp1r*^{-/-} motoneurons. We performed Pabp immunoprecipitation and eIF4G immunoblot analysis on compartmentalized wildtype motoneurons. We were able to detect an association between eIF4G and Pabp in axons but this required 24 embryos for a single experiment. Since *Hnrnp1r*^{-/-} mice are derived from crosses of

Hnrnpr^{-/-} mice, representing 25% of the offspring, it is therefore not feasible to perform this experiment on compartmentalized *Hnrnpr*^{-/-} motoneurons.

9. Taking together the data in this study, the model in Fig. 5g, may not be accurate, although this reviewer understood that this cartoon is intended to summarize the results up to Fig. 5. Given the hnRNP R enhances Ogt-mediated eIF4G O-GlcNAc for PABP interaction and translation activation, placing the hnRNP R as a part of the complex (not highlighting the above points) may be confusing to readers. The authors should consider to re-draw the model to fit the current results.

Author's response: We agree with the reviewer and have removed the model from Fig. 5 (now Fig. 6 in the revised manuscript). Instead, we present a model summarizing all findings as Fig. 10.

10. Once the experimental setup in Fig. 6f could monitor O-GlycNAcylation of eIF4G, the data showed the TMG treatment also increased O-GlycNAcylation of eIF4G in soma as well. This reviewer wonders why this did not lead to translation activation in soma (Fig. 6g).

Author's response: According to our data in Fig. 6f (now Fig. 8e,f in the revised manuscript), the O-GlcNAcylation of eIF4G is not significantly increased in the somatodendritic compartment of *Hnrnpr*^{-/-} motoneurons upon TMG treatment ($P = 0.4126$). This result therefore is in line with our finding that TMG treatment does not affect translation in the somatodendritic compartment of *Hnrnpr*^{-/-} motoneurons (Fig. 8i,j in the revised manuscript).

Minor points:

1. At lines 219-219, given the data provided in Fig. 4a-g, it was quite hard to specify that hnRNP R's regulatory point lies at translation initiation. The authors should tone down this claim.

Author's response: We followed the reviewer's advice and we have updated our manuscript as follows:

"Together, these results point towards the possibility that hnRNP R regulates translation at the initiation stage."

2. Regarding Fig. 5a, it would be straightforward to speculate that the hnRNP R-mediated eIF4G-PABP interaction enhancement is based on the O-GlcNAcylation of eIF4G. The authors should mention this interpretation in the text.

Author's response: We very much agree with the reviewer that O-GlcNAcylation of eIF4G contributes to the enhanced eIF4G-Pabp interaction in axons of motoneurons, and that this is dependent on hnRNP R. Therefore, we added the following to the Discussion:

"This suggests that hnRNP R is required for Ogt activity in axons, and thus for translation through O-GlcNAcylation of eIF4G, which increases its binding to Pabp."

3. The authors may consider moving Figures 1a, b, d, e, and f (concerning KO validation) to the Extended Figure section.

Author's response: We have followed the reviewer's suggestion and moved the figures mentioned to Supplementary Fig. 1 as following:

Fig. 1

Extended Data Fig. 1

4. The custom gene ontology category of “cytoskeleton” used in Fig. 2h-i, should be explained in detail in the method section.

Author’s response: As suggested by the reviewer, we added the list of custom cytoskeletal proteins as Supplementary Table 4 and added the following sentence to the Results section in order to explain the rationale behind this selection:

“Additionally, while we did not observe a general downregulation of proteins with cytoskeletal functions, we identified several proteins with known cytoskeletal roles in axon development and maintenance such as *Stmn2*, *Stmn4*, *Tuba1a*, *Tubb2a*, and *Tubb4a* that were reduced in axons but not somatodendritic regions of *Hnrnp1r*^{-/-} motoneurons (Fig. 2h,i ‘custom cytoskeleton’; Supplementary Table 4).”

5. It should be helpful to denote the “Ogt” in the left panel of Fig. 6a (Somatodendritic panel). Related to this, it would be informative to perform IP-western experiments between Ogt and hnRNP R in both somatodendritic and axon fractions.

Author’s response: According to our data, Ogt is not enriched in the hnRNP R immunoprecipitate from the somatodendritic compartment ($P = 0.153$). Therefore, Ogt is a major interactor of hnRNP R only in axons. To clarify this, we now labelled “Ogt” in the hnRNP R interactome data from the somatodendritic compartment as suggested by the reviewer (Fig. 7a in the revised manuscript). We also performed the reverse experiment, immunoprecipitating Ogt from axonal and somatodendritic compartments followed by mass spectrometry analysis (Fig. 7c in the revised manuscript). Likewise, we found that hnRNP R was enriched in the Ogt immunoprecipitate only from axons. We labelled both “hnRNP R” and “Ogt” in the corresponding volcano plots.

As suggested by the reviewer, we have performed hnRNP R immunoprecipitation in somatodendritic and axonal compartments of motoneurons followed by western blot to validate the interaction between Ogt and hnRNP R in axons. We observed a strong interaction of Ogt with hnRNP R in axons but not in the somatodendritic compartment (Fig. 7f in the revised manuscript).

Fig. 7

6. Typo in Fig. 1g:

“DAPI/Tubulin/hnRPN R” should be “DAPI/Tubulin/hnRNP R”

Author’s response: We thank the reviewer for identifying this typo. We now have updated the Fig.1g in the revised manuscript.

7. For Fig. 2h, the authors should consider a cumulative fraction plot or box plot for better visualization and statistical analysis.

Author’s response: We followed the reviewer’s suggestion and presented the data in Fig. 2h as box plot, including a statistical analysis of the data as follows:

Fig. 2h

8. Regarding Fig. 6a, the authors should directly compare the interaction strength of each protein found in somatodendritic and axon fractions. This reviewer wonders whether a general correlation could be found between the two data sets.

Author's response: As suggested by the reviewer, we compared the proteins enriched ($P < 0.05$) in the hnRNP R immunoprecipitate from the somatodendritic compartment with those from the axonal compartment. We found that 15 proteins associated with hnRNP R in the somatodendritic compartment also interacted with hnRNP R in the axonal compartment. We included this information in the revised manuscript in Fig. 7b as follows:

“Among hnRNP R interactors, only 15 proteins associated with hnRNP R in the somatodendritic compartment also interacted with hnRNP R in the axonal compartment further indicating that axonal hnRNP R complexes have a different composition compared to somatodendritic hnRNP R complexes (Fig. 7b).”

Fig. 7b

Reviewer #3 (Remarks to the Author):

In the manuscript entitled “hnRNP R promotes O-GlcNAcylation of eIF4G to facilitate axonal protein synthesis”, Zare and colleagues investigate the role of the RNA-binding protein hnRNP R in local axonal protein synthesis and its effect on motoneuron function. To this end, they generated a new *Hnrnpr* knockout mouse, providing them the opportunity to investigate functions in vivo. Using this approach, they validated their previous findings that hnRNP R depletion affects axonal outgrowth. Using a range of imaging and ‘omics’ methods, the authors uncover that hnRNP R deficiency leads to perturbed local protein synthesis and an altered axonal proteome. Furthermore, the authors uncover O-GlcNAcylation as a mechanism by which translation initiation is likely affected by hnRNP R via its regulation of the O-GlcNAc transferase. Though experiments in living mice are limited to a few initial readouts, they confirm that the axon outgrowth defects observed in the current and previous studies indeed lead to motor defects. In summary, the presented data nicely validates and expands upon the authors' previous publications. The experiments are generally well-designed, the manuscripts well-written, and the data presented in clear graphs. Some relatively minor inquiries remain.

Author’s response: We appreciate the reviewer’s positive assessment of our manuscript.

- For Fig. 1h: could the authors please specify how survival was assessed in the methods?

Author’s response: We have adjusted the Methods section explaining our analysis in more detail in the section “Survival analysis” as follows:

“For analysis of motoneuron survival, *Hnrnpr*^{-/-} and +/+ motoneurons were cultured on laminin-111-coated glass coverslips for 6 DIV. Number of surviving neurons that were identified by morphological appearance of the cell body and intact neurites were quantified manually in defined areas on each coverslip on DIV 2, 4, and 6.”

- The authors should specify not only the number of biological replicates but also the number of axons quantified per replicate (e.g. Fig. 1j) in the figure legend for all relevant graphs.

Author’s response: We have now included the number of individual motoneurons that were analyzed per condition in the figure legends for all figures. For axon length measurements, only

the length of the longest axonal branch was measured in each individual motoneuron, as outlined in the methods section Image acquisition and data analysis.

- Though it is intuitive, in Fig. 2i could the authors state what is encoded by the size of the circle in the figure legend.

Author's response: The size of the circle represents the number of proteins. We now included this information in the figure legend as follows:

“Enrichment scores of protein subgroups in axons of *Hnrnpr*^{-/-} relative to +/+ motoneurons. The size of the circles represents the number of proteins.”

- The title in line 166 states proteome alterations are not due to mRNA transport defects. Though this is a likely interpretation, this is not directly shown by the data or further specified in the following paragraphs. Therefore, I would ask the authors to adapt the title and elaborate on this topic rather in the discussion.

Author's response: We followed the reviewer's advice and adjusted the title, Results section, and Discussion in the revised manuscript as follows:

Figure title:

“Axonal proteome alterations of hnRNP R knockout motoneurons are not due to altered levels of axonal mRNA”

Results section:

“In order to assess whether dysregulation of axonal proteins in *Hnrnpr* knockout motoneurons is due to altered axonal levels of encoding transcripts, we investigated changes in the somatodendritic and axonal mRNA levels in compartmentalized *Hnrnpr*^{-/-} and +/+ motoneurons by RNA sequencing (RNA-seq).”

Discussion section:

“To shed light on the mechanisms underlying hnRNP R's role in axonal translation regulation, we performed mass spectrometry on axonal lysates derived from *Hnrnpr*^{-/-} and +/+ mouse motoneurons, detecting widespread changes in protein abundance in hnRNP R-deficient axons. Additionally, our RNA-seq data from somatodendritic and axonal compartments of hnRNP R knockout motoneurons revealed that these axonal proteome alterations are not simply due to

changes in the axonal abundance of the encoding transcripts. Therefore, while hnRNP R has functions in axonal mRNA translocation (Glinka et al. Hum Mol Genet doi: 10.1093/hmg/ddq073.), it exerts functionally separate roles in local translation.”

Is there an overlap between targets regulated in axons and the somatodendritic compartment?

Author’s response: In order to address this point, we cross-compared our RNA-seq data from axonal and somatodendritic compartments. We found that among upregulated axonal transcripts, 5 were also upregulated in the somatodendritic compartment, and, among the downregulated axonal transcripts, 3 were also downregulated in the somatodendritic compartment of *Hnrnr^{-/-}* motoneurons. We included this information in the Results section of the revised manuscripts as follows:

“Using a less stringent cutoff of $P < 0.05$, we identified 1,195 dysregulated axonal transcripts, of which 412 transcripts were upregulated, 783 transcripts were downregulated, and 198 dysregulated somatodendritic transcripts, of which 112 were upregulated and 86 were downregulated (Fig. 4c). Among these, 5 and 3 transcripts were up- or downregulated, respectively, in both somatodendritic and axonal compartments of *Hnrnr^{-/-}* motoneurons (Figure. 4d).“

Out of curiosity, would the observation that downregulated targets in axons contain more iCLIP sites (Fig. 3d) also be true for the somatodendritic compartment?

Author’s response: We thank the reviewer for bringing up this point. Our data show that, in the somatodendritic compartment, upregulated transcripts also contain less iCLIP sites. We have updated the manuscript as follows:

“We observed that transcripts downregulated in axonal compartments of *Hnrnp1*^{-/-} motoneurons contain more hnRNP R iCLIP hits whereas upregulated transcripts in both somatodendritic and axonal compartments contain less hnRNP R iCLIP hits compared to unchanged transcripts (Fig. 4e).”

Fig. 4e

- The authors should clearly state in the text what each figure shows. For instance, explain that Extended data 3a is the control for the mentioned data.

Author’s response: We followed the reviewer’s suggestion and we included this information in the revised manuscript, for example:

“We observed that the hnRNP R-eIF2 α PLA signal was detectable not only in the cytosol of the somata but also in axons and growth cones of motoneurons (Fig. 5h). No PLA signals were detectable when either hnRNP R or eIF2 α antibody was omitted indicating specificity of the procedure (Supplementary Fig. 3a).”

- In Fig. 6a, is Ogt also enriched in the somatodendritic compartment? Could the authors indicate Ogt in the somatodendritic compartment?

Author’s response: According to our data, Ogt is not enriched in the hnRNP R immunoprecipitate from the somatodendritic compartment ($P = 0.153$). Therefore, Ogt is a major interactor of hnRNP R only in axons. To clarify this, we now labelled “Ogt” in the hnRNP R

interactome data from the somatodendritic compartment as suggested by the reviewer (Fig. 7a in the revised manuscript). We also performed the reverse experiment, immunoprecipitating Ogt from axonal and somatodendritic compartments followed by mass spectrometry analysis (Fig. 7c in the revised manuscript). Likewise, we found that hnRNP R was enriched in the Ogt immunoprecipitate only from axons. We labelled both “hnRNP R” and “Ogt” in the corresponding volcano plots.

Fig. 7c

- It is indeed interesting that TMG treatment can restore O-GlcNAc levels (Extended Data Fig. 4). However, would it not be more informative to attempt to rescue O-GlcNAc levels by reintroducing hnRNP R. This would validate the direct involvement of hnRNP R which is the final interpretation of this paragraph.

Author’s response: We thank the reviewer for this excellent suggestion. To test this possibility, we generated a lentiviral construct expressing an hnRNP R-EGFP fusion protein in order to reintroduce hnRNP R into *Hnrnp1r*^{-/-} motoneurons. As a result, we found that expression of hnRNP R-EGFP could restore O-GlcNAcylation of eIF4G and also rescue the axon growth defect of *Hnrnp1r*^{-/-} motoneurons. We have included this data in the revised manuscript as following:

“To further validate that the reduction of axonal O-GlcNAcylation of eIF4G in *Hnrnp1r*^{-/-} motoneurons is the consequence of hnRNP R loss, we transduced motoneurons cultured from *Hnrnp1r*^{-/-} and +/+ mice with a lentiviral construct expressing an hnRNP R-EGFP fusion protein. Expression of hnRNP R-EGFP could restore the level of O-GlcNAcylation of axonal eIF4G in hnRNP R-depleted motoneurons (Supplementary Fig. 5b, c). Similar to TMG treatment, reintroducing hnRNP R could rescue the axon elongation defect observed in *Hnrnp1r*^{-/-} motoneurons (Supplementary Fig. 5f, g).”

Supplementary Fig. 5

• In line 341 in the discussion, the author state that translation was assessed in the NMJ. Could the author point out this data, as we might have missed it in the manuscript.

Author’s response: We apologize for the confusion; we did not include any data on translation at NMJs in the manuscript. We therefore changed the respective sentence in the Discussion as following:

“We observed that depletion of hnRNP R in motoneurons leads to markedly decreased protein synthesis exclusively in axons and axon terminals but not the somatodendritic compartment in cultured motoneurons of *Hnmp1^{-/-}* mice.”

- Perhaps the authors could mention/discuss the relevance of Ptbp2 for hnRNP R and the current data as discovered in their previous publication.

Author's response: We thank the reviewer for pointing this out. We included the information and expanded the Discussion in the revised manuscript as follows:

“RBPs have previously been shown to regulate translation by facilitating the formation of pre-initiation complexes, or by strengthening or repressing interactions between the mRNA and the ribosome. For example, we recently showed that the neuronal RBP Ptbp2 promotes axonal translation of *Hnrnpr* mRNA by facilitating its association with translating ribosomes.”

And:

“Interestingly, hnRNP R itself has recently been shown to be locally synthesized from axonal *Hnrnpr* transcripts in a manner dependent on the RBP Ptbp2, indicating cross-regulation of axonal RBPs to fine-tune axonal protein-synthesis.”

- The figures are partially somewhat crowded, and specifically microscopy images are quite small, which may be even more drastic in a final layout. Perhaps the authors could adjust image size, text size, etc. so the data can be easily seen in a final layout.

Author's response: We followed the reviewer's suggestion and changed figures as follows in order to improve clarity:

For Fig. 1, we moved several images regarding the validation of the knockout mice to the Supplement as new Supplementary Data Fig. 1. Reviewer #2 also recommended this.

For the proteomics data in Fig. 2h, we replaced the volcano plots with box plots, which also allow quantification and statistical analysis.

We split Fig. 6 in two and moved the data on eIF4G O-GlcNAcylation, Puromycin incorporation, and TMG treatment to a new Fig. 8. This way, there is no overcrowding of Fig. 6 (new Fig. 7 in the revised manuscript) anymore.

We removed superfluous dotted lines from microscopy images to enhance their clarity.

REVIEWER COMMENTS

Reviewer #1 (Remarks to the Author):

This is a revised manuscript which I had reviewed previously. The authors added additional data and provide new analysis, which clarified all my previous concerns.

Reviewer #2 (Remarks to the Author):

This reviewer appreciated the authors' efforts in improving this manuscript. However, this reviewer still has concerns to be answered before the publication of this manuscript.

Major comments

1.

The eIF4A band at the bottom of new Fig. 8d is inappropriately cropped, judged by the signal from input lanes (probably due to the membrane cut on the protein band). This reviewer recommends revising this point.

2.

Related to point 1, the same IP-Western blot experiment using hnRNP R KO should be considered. This experiment further ensured that eIF4A O-GlcNAcylation is not relevant.

3.

In either case, this manuscript still did not provide direct evidence that the O-GlcNAcylation on "eIF4G" explains the axon local translation they observed. Still, there are enough possibilities to be explained by other unknown proteins with O-GlcNAcylation. Thus, authors should consider to tone down the description (including the title of the paper).

Minor comments

1.

The descriptions in the supplementary tables were not user-friendly. For example, the column names of Supplementary Table 1, 9, and 10 (e.g.

E:\Jakob\33G2_A\20240310_OA1_Evo12_31min_JaBa_SA_J141_33G2_DIA_8UL_sample01.raw, [1] 20210920_TIMS05_JaBa_SA_33EXP2_120min_Py3diaPASEF100ms_sample01_C1_1_5487.d.PG.Quantity) was quite complicated. Please just include the simplified version of the information in the supplementary files (conditions, column names, etc..) for the readers.

3. For new Fig. 8i-j, the method for the quantification of puromycylation was missing. Visually, the data of +TMG looked brighter than the control in Fig. 8i. However, the quantification shown in Fig. 8j was similar. This was counterintuitive.

Reviewer #3 (Remarks to the Author):

The authors did a very careful and thorough approach to successfully address the points raised. In all cases, they improved/clarified the study. Here, I want to particularly mention the rescue approach that hnRNP R-eGFP restores O-GlcNAcylation, this further boosts the confidence in the presented data. I therefore strongly suggest publication. Congratulations to the authors for a fine job.

Overall, by reading the entire rebuttal, my got the impression that the authors really addressed most points experimentally, if feasible/possible. All data shown are logical, trustable and of high quality. I appreciate the time and effort to make this study better. They succeeded...

Reviewer #1 (Remarks to the Author):

This is a revised manuscript which I had reviewed previously. The authors added additional data and provide new analysis, which clarified all my previous concerns.

Reviewer #2 (Remarks to the Author):

This reviewer appreciated the authors' efforts in improving this manuscript. However, this reviewer still has concerns to be answered before the publication of this manuscript.

Major comments

1. The eIF4A band at the bottom of new Fig. 8d is inappropriately cropped, judged by the signal from input lanes (probably due to the membrane cut on the protein band). This reviewer recommends revising this point.

Author's response: We followed the reviewer's suggestion and updated Fig. 8d by providing larger sections of the eIF4A blots. In addition to the eIF4A bands, the heavy chain of the IP antibody at ~55 kDa is visible, which we marked with asterisks. We have also provided an uncropped Western blot below and included it in the source data.

2. Related to point 1, the same IP-Western blot experiment using hnRNP R KO should be considered. This experiment further ensured that eIF4A O-GlcNAcylation is not relevant.

Author's response: For the RL2 immunoprecipitation followed by proteomics, we have used 6 hnRNP R KO embryos for each replicate and the Orbitrap Astral (Thermo Fisher Scientific) mass spectrometer for high-sensitivity measurements. However, for the RL2 IP Western blot (Fig. 8d) we used wildtype 40 embryos to obtain a visible eIF4A signal for axonal IPs. Since *Hnrnp*^{-/-} mice are derived from crosses of *Hnrnp*^{+/-} mice, representing 25% of the offspring, obtaining 40 hnRNP R KO embryos at the same time is technically not feasible.

3. In either case, this manuscript still did not provide direct evidence that the O-GlcNAcylation on "eIF4G" explains the axon local translation they observed. Still, there are enough possibilities to be explained by other unknown proteins with O-GlcNAcylation. Thus, authors should consider to tone down the description (including the title of the paper).

Author's response: According to the reviewer's suggestion, we now changed the title and updated the Abstract, Results, and Discussion in the revised manuscript as follows:

Title: "hnRNP R promotes O-GlcNAcylation of eIF4G and facilitates axonal protein synthesis

Abstract: "In axons, hnRNP R is a component of translation initiation complexes and, through interaction with O-linked β -N-acetylglucosamine (O-GlcNAc) transferase (Ogt), modulates O-GlcNAcylation of eIF4G."

Results: "Thus, while we cannot exclude the possibility that O-GlcNAcylation of additional proteins regulating translation is dependent on hnRNP R, our results indicate that hnRNP R positively regulates eIF4G activity and, thus, formation of initiation complexes by stimulating Ogt."

Discussion: "Thiamet-G treatment could rescue axonal O-GlcNAcylation of eIF4G and stimulate eIF4G binding to Pabp, accompanied by restored axonal protein synthesis, in hnRNP R-deficient motoneurons. This suggests that hnRNP R is required for Ogt-mediated O-GlcNAcylation of eIF4G, which increases its binding to Pabp."

Minor comments

1. The descriptions in the supplementary tables were not user-friendly. For example, the column names of Supplementary Table 1, 9, and 10 (e.g. E:\Jakob\33G2_A\20240310_OA1_Evo12_31min_JaBa_SA_J141_33G2_DIA_8UL_sample01.raw, [1] 20210920_TIMS05_JaBa_SA_33EXP2_120min_Py3diaPASEF100ms_sample01_C1_1_5487.d.PG.Quantity) was quite complicated. Please just include the simplified version of the information in the supplementary files (conditions, column names, etc..) for the readers.

Author's response: We followed the reviewer's advice and we have updated the sample names in the supplementary tables.

2. For new Fig. 8i-j, the method for the quantification of puromycylation was missing. Visually, the data of +TMG looked brighter than the control in Fig. 8i. However, the quantification shown in Fig. 8j was similar. This was counterintuitive.

Author's response: We have now included the quantification of puromycylation in the method section of the revised manuscript as follows:

"For intensity measurements of puromycin, raw images were projected using ImageJ and mean gray values were measured after background subtraction. Puromycin intensity was measured in somata and 20 μ m-long proximal and distal regions of axons."

For the axonal puromycin intensity measurements in wildtype motoneurons, we noticed a slight increase in puromycin intensity upon TMG treatment, which, however, was not significant (Fig. 8j, proximal axons).

Reviewer #3 (Remarks to the Author):

The authors did a very careful and thorough approach to successfully address the points raised. In all cases, they improved/clarified the study. Here, I want to particularly mention the rescue approach that hmRNP R-eGFP restores O-GlcNAcylation, this further boosts the confidence in the presented data. I therefore strongly suggest publication. Congratulations to the authors for a fine job.

Overall, by reading the entire rebuttal, my got the impression that the authors really addressed most points experimentally, if feasible/possible. All data shown are logical, trustable and of high quality. I appreciate the time and effort to make this study better. They succeeded...